# Proenkephalin-A secreted by renal proximal tubules functions as a brake in kidney regeneration

Chi Liu [1,3] ✉, Xiaoliang Liu [1,3], Zhongwei He[1,3], Jiangping Zhang[1,3], Xiaoqin Tan[1], Wenmin Yang[1], Yunfeng Zhang[1], Ting Yu[2], Shuyi Liao[1], Lu Dai[1], Zhi Xu[2], Furong Li[1], Yinghui Huang [1] ✉ & Jinghong Zhao [1] ✉

Organ regeneration necessitates precise coordination of accelerators and brakes to restore organ function. However, the mechanisms underlying this intricate molecular crosstalk remain elusive. In this study, the level of proenkephalin-A (PENK-A), expressed by renal proximal tubular epithelial cells, decreases significantly with the loss of renal proximal tubules and increased at the termination phase of zebrafish kidney regeneration. Notably, this change contrasts with the role of hydrogen peroxide ($H_2O_2$), which acts as an accelerator in kidney regeneration. Through experiments with *penka* mutants and pharmaceutical treatments, we demonstrate that PENK-A inhibits $H_2O_2$ production in a dose-dependent manner, suggesting its involvement in regulating the rate and termination of regeneration. Furthermore, $H_2O_2$ influences the expression of *tcf21*, a vital factor in the formation of renal progenitor cell aggregates, by remodeling H3K4me3 in renal cells. Overall, our findings highlight the regulatory role of PENK-A as a brake in kidney regeneration.

Multiple organs can recover their original functions through regeneration after injury. For example, the human kidneys can be partially repaired after renal tubular injury, as surviving tubular epithelial cells can repair various injuries via proliferation and migration[1]. The human liver can recover its original size and function following partial hepatectomy (PHx)[2]. During organ regeneration, many kinds of accelerators promote regeneration[3]. However, excessive activation of regeneration accelerators can lead to disease[4], highlighting the importance of precise regulation. Thus, the body must finely control the levels of these accelerators to ensure successful organ regeneration.

Considerable efforts have been dedicated to investigating the mechanisms triggering organ regeneration. Calcium ($Ca^{2+}$)

signaling[5,6], reactive oxygen species (ROS)[7–9], inflammation[10–13], and nerve-related factors[14,15] have been demonstrated to act as accelerators in the regeneration of various organs. However, the understanding of regeneration "brakes" remains limited[16]. For instance, integrin-linked kinase (ILK) has been established as a termination signal in liver regeneration. Mice with liver-specific ILK ablation exhibit 58% larger liver following PHx than controls[17]. Glypican 3[18], C/EBPα[19], and HNF4α[20] are also related to the suppression of liver regeneration. In the case of heart regeneration in adult mice, the Hippo pathway serves as a brake; blocking Hippo signaling components enables adult cardiomyocyte renewal after myocardial infarction[21]. Nonetheless, the molecular brake involved in kidney regeneration remains elusive.

[1]Department of Nephrology, the Key Laboratory for the Prevention and Treatment of Chronic Kidney Disease of Chongqing, Chongqing Clinical Research Center of Kidney and Urology Diseases, Xinqiao Hospital, Army Medical University (Third Military Medical University), 400037 Chongqing, P.R. China. [2]Department of Respiratory Medicine, Xinqiao Hospital, Army Medical University (Third Military Medical University), 400037 Chongqing, P.R. China. [3]These authors contributed equally: Chi Liu, Xiaoliang Liu, Zhongwei He, Jiangping Zhang. ✉e-mail: chiliu@tmmu.edu.cn; ikkyhuang@163.com; zhaojh@tmmu.edu.cn

Zebrafish kidney development consists of two stages, namely the pronephros and mesonephros stages. During the embryonic stage, the zebrafish pronephros consists of two nephrons, whereas the adult zebrafish mesonephros comprises approximately 500 nephrons[22]. Unlike mammals, zebrafish possess a robust capacity for kidney regeneration. Intraperitoneal injection of excessive gentamicin (Gent) in adult zebrafish leads to apoptosis of renal tubular epithelial cells, resulting in acute kidney injury (AKI)[22–24]. However, zebrafish can rapidly regenerate a significant number of new nephrons to repair kidney damage[22]. *lhx1a* positive (*lhx1a*⁺) renal progenitor cells (RPCs) in the renal medulla aggregate at the distal segments of the renal tubules to produce RPC aggregates (RPCAs) after AKI. Subsequently, these cell aggregates can differentiate into mature nephrons[22,25]. Therefore, the zebrafish is a promising animal model for investigating kidney regeneration. While some mechanisms of kidney regeneration have been elucidated in this model[25–28], further clarification is needed regarding the molecular mechanism that acts as a brake during kidney regeneration.

Previous investigations from our laboratory have demonstrated that the zebrafish kidney exhibits a pronounced ability to generate substantial quantities of hydrogen peroxide ($H_2O_2$) in response to injury[29]. In the initial stage of kidney regeneration, suppressing $H_2O_2$ production effectively diminishes the formation of *lhx1a*⁺ RPCAs and impedes the onset of kidney regeneration[29]. Thus, $H_2O_2$ functions as an initial signaling molecule in kidney regeneration. However, the effect of $H_2O_2$ on the kidney is dual-sided. High concentrations of $H_2O_2$ can cause renal injury and fibrosis[30,31]. Consequently, the levels of $H_2O_2$ in the kidney necessitate meticulous regulation. However, the precise mechanism underlying this process remains incompletely understood.

Human proenkephalin (PENK) is an endogenous opioid polypeptide hormone. Upon proteolytic cleavage, PENK generates enkephalin peptides, predominantly Met-enkephalin (Met-ENK) and, to a lesser extent, Leu-enkephalin (Leu-ENK)[32,33]. These enkephalin peptides can activate opioid receptors, influencing physiological functions such as pain perception and stress responses[34]. PENK has emerged as a potential biomarker for assessing kidney function, showing a robust negative correlation with estimated glomerular filtration rate (eGFR), thus serving as a kidney biomarker of glomerular function[32,33]. In addition, increased plasma PENK concentrations have also been associated with long-term outcomes in AKI and cardiac diseases[35]. In addition, PENK is reportedly expressed in the renal tubules of rats[36], suggesting that intrarenal PENK may participate in normal kidney function and kidney diseases. However, research on these topics is still limited.

In the present study, we observed a significant correlation between the expression of *penka* (an ortholog of human *PENK*) in zebrafish proximal tubule epithelial cells (PTECs) and the state of kidney regeneration. Specifically, *penka* levels decreased during the loss of proximal tubules (PTs) and increased upon their recovery at the end of the kidney regeneration process. Interestingly, these changes in *penka* levels were in contrast to the levels of $H_2O_2$. Through the use of *penka* mutants and antagonists, we demonstrated that inhibiting the PENK-A signaling pathway led to increased $H_2O_2$ production and accelerated kidney regeneration. Conversely, activation of PENK-A reduced $H_2O_2$ production and resulted in earlier termination of kidney regeneration. Moreover, our findings revealed that transcription factor 21 (*tcf21*), a critical factor in RPCA formation, was regulated by $H_2O_2$. $H_2O_2$ promoted the expression of *tcf21* by remodeling trimethylation at the 4th lysine residue of the histone H3 protein (H3K4me3) in renal cells. Collectively, our results highlight the role of PENK-A as a negative feedback regulator in kidney regeneration.

## Results

### Specific expression of *penka* in PTECs

AKI is defined as sudden damage to a significant number of nephrons (the functional units of the kidneys)[37]. However, it is not clear whether nephron cells affect their own regeneration. To answer this question, we reanalyzed the previously published single-cell RNA sequencing (scRNA-seq) data of adult zebrafish kidneys[38]. Our analysis revealed that *penka* was specifically expressed in PTECs (Fig. 1a). The zebrafish *penka* gene encodes an enkephalin precursor that undergoes processing to produce four Met-ENKs, as well as one Met-enkephalin-Ile and one Met-enkephalin-Asp[39]. To confirm the expression pattern of *penka*, we initially utilized immunofluorescence staining for a marker for renal tubules, Pax2a[40], combined with *penka* fluorescence in situ hybridization (FISH). We observed that *penka* is likely expressed in the PTs (Fig. 1b). PTs in zebrafish consist of proximal convoluted tubules (PCTs) and proximal straight tubules (PSTs)[41]. By using FISH with markers for PCT and PST[42], namely, *slc20a1a* and *trmp7*, respectively, we observed that the Met-ENK immunofluorescence signal co-localized with the signals of *slc20a1a* and *trmp7* (Fig. 1c), confirming the expression of *penka* in the PTs of adult zebrafish kidneys. Furthermore, we investigated *PENK* expression in normal human kidney samples and samples from patients with AKI. Through FISH analysis, we observed high *PENK* expression in the human PTs (Fig. 1d), as indicated by colocalization with lotus tetragonolobus lectin (LTL, a marker for PTs)[43] staining. Importantly, in AKI patients, who experienced loss of PTs, *PENK* expression was significantly decreased (Fig. 1d). These findings suggest a potential role for *PENK* in AKI. Therefore, further exploration of PENK-A's involvement in kidney injury using the zebrafish model holds promise for providing valuable insights for clinical treatment.

Next, we examined the expression of *penka* during kidney regeneration in an adult zebrafish AKI model. To establish the model, we intraperitoneally injected Gent (2.7 µg/µL, 20 µL per fish) into adult zebrafish[22,25]. Gent can accumulate in the lysosomes of PTECs, and high doses can induce PTEC apoptosis, leading to AKI[44]. The kidney injuries mentioned subsequently in this study were all induced using this established adult zebrafish AKI model. Subsequently, we used reverse transcription polymerase chain reaction (RT-PCR) (Fig. 1e) and quantitative Real-time PCR (qRT-PCR) (Fig. 1f) to examine the changes in *penka* expression during the kidney regeneration process in this model. Our results showed that *penka* expression in kidneys decreased initially during regeneration, reached its lowest level at 7 days post-injury (7 dpi), and then increased from 9 dpi, returning to normal levels by 15 dpi (Fig. 1e, f). Furthermore, we quantified the number of nephrons in the *Tg(gtshβ:GFP)*[45] kidneys, which involved labeling of PTs with GFP, at 0, 3, 5, 7, 9, and 11 days after injury (Supplementary Fig. S1a). By quantification, we found that the number of PTs (corresponding to the number of nephrons) exhibited a pattern of change similar to that of the expression of *penka* (Supplementary Fig. S1b). Based on this correspondence, we speculate that changes in *penka* expression may play a crucial role in kidney regeneration.

### Deficiency of PENK-A accelerates kidney regeneration

To investigate the impact of *penka* on kidney regeneration, two *penka* mutants were generated using CRISPR/Cas9-induced gene knockout. The two mutation types identified were *penka*⁻¹⁺²⁴ (a 1-bp deletion and a 24-bp insertion) and *penka*⁻⁸ (an 8-bp deletion) (Supplementary Fig. S2a). Since there was no significant disparity in the regeneration phenotype observed between the two mutants, we primarily employed the *penka*⁻¹⁺²⁴ mutant for subsequent experiments. Regarding the embryonic and adult stages, including the size and structure of the adult kidneys, there were no discernible differences between the *penka*⁻¹⁺²⁴ mutants and the wild type (WT) (Supplementary Fig. S2b, c). In addition, an off-target analysis was conducted on this strain. We utilized CRISPRScan for off-target prediction at the target site and selected the top five potential off-target sites for sequencing analysis[46,47]. No off-target effects were observed at these five sites in the mutants compared to the WT (Supplementary Fig. S2d, e). Regarding the expression of *lhx1a*, a marker gene of RPCAs[22,25], the

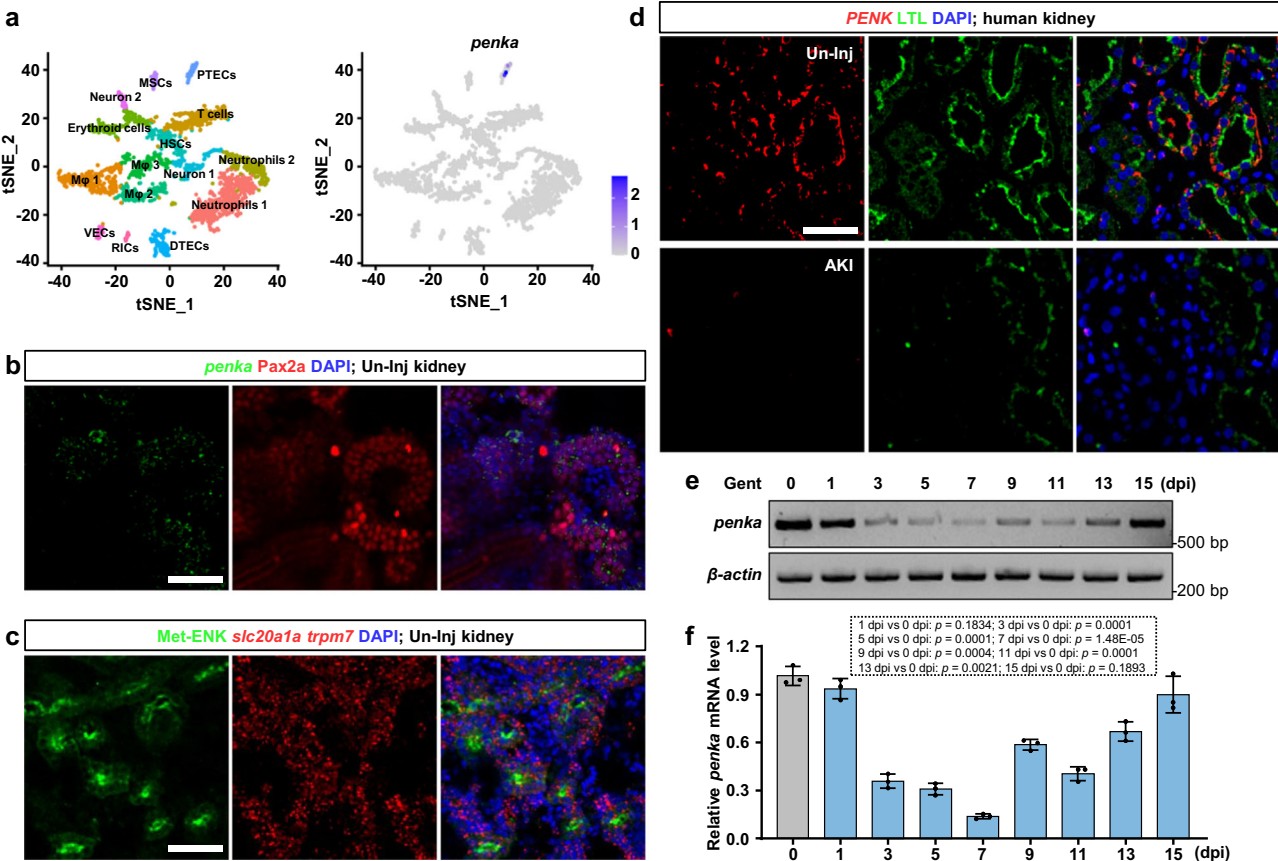

**Fig. 1 | The expression patterns of zebrafish *penka* and human *PENK* in the kidneys. a** scRNA-seq analysis revealed that *penka* was specifically expressed in zebrafish PTECs. t-SNE plots showing zebrafish kidney cell clusters and the expression of *penka*. VECs, vascular endothelial cells; DTECs, distal tubular epithelial cells; Mφ, macrophages; HSCs, hematopoietic stem cells; MSCs, mucin-secreting cells; and RICs, renal interstitial cells. **b** Confocal images showing double labeling of FISH-*penka* and anti-Pax2a in un-injured (Un-Inj) adult zebrafish kidney sections (*n* = 3). **c** Confocal images showing triple labeling of FISH-*slc20a1a*, FISH-*trpm7*, and anti-Met-ENK in Un-Inj adult zebrafish kidney sections. The Met-ENK signal co-localized with the signals of *slc20a1a* and *trmp7*, which are markers of PCT

and PST, respectively (*n* = 3). **d** Confocal images of combined FISH-*PENK* and LTL staining in kidney sections of patients with AKI and patients with no detectable lesions (Un-Inj). Human *PENK* was expressed in PTs and downregulated after AKI (*n* = 3). **e**, **f** RT-PCR (**e**) and qRT-PCR (**f**) analyses of *penka* in zebrafish kidneys during Gent-induced AKI (*n* = 3). *penka* expression was decreased by 1 dpi and reached its lowest level at 7 dpi and returned to its un-injured level at 15 dpi. The data in (**f**) were analyzed by two-sided t-test and are presented as mean values ± SD. Scale bars in (**b**), (**c**), and (**d**), 50 μm. Source data are provided as a Source data file.

*penka*[−/−24] mutant kidneys exhibited high expression levels at 5 dpi, as shown by RT-PCR analyses, whereas the control kidneys reached this level at 7 dpi (Fig. 2a, b). Furthermore, whole-mount in situ hybridization (WISH) of *lhx1a* revealed that, compared to that in the WT counterparts, the number of RPCAs in the mutant adult kidneys significantly increased at 5 dpi, while it decreased at 9 dpi (Fig. 2c, d). These findings indicate that loss of the *penka* gene can accelerate kidney regeneration. In addition, we performed knockdown experiments using *penka* vivo-morpholino (MO) (100 μM, 10 μL per fish), which effectively interferes with the processing of *penka* mRNA (Supplementary Fig. S3a, b). After intraperitoneal injection of *penka* vivo-MO, we observed a significant increase in the number of RPCAs at 5 dpi and 7 dpi, as determined by qRT-PCR and WISH analysis of *lhx1a* expression (Supplementary Fig. S3c–e). Subsequently, we intraperitoneally injected the PENK-A antagonist naloxone methiodide[48] (NAL-M, 2.0 μM, 10 μL per fish) at 2, 4, 6, and 8 dpi. RT-PCR and WISH analysis of the kidneys treated with NAL-M revealed a significant increase in the number of *lhx1a*[+] RPCAs compared to that in the control group at 5 dpi. At 7 dpi, there was a slight increase in the number of RPCAs, but interestingly, at 9 dpi, the number decreased (Fig. 2e–g). These findings indicate that *penka* acts as a negative regulator of kidney regeneration and that deletion of *penka* accelerates kidney regeneration.

## PENK-A acts as a brake in kidney regeneration

When the PENK-A agonist Met-ENK[49] (100 μM, 10 μL per fish) and tramadol[50] (TRAM, 16 μM, 10 μL per fish) were administered via intraperitoneal injection at 2, 4, and 6 dpi, a noticeable decrease in the expression of *lhx1a* and a reduction in the number of RPCAs were observed in the treated group at 7 dpi through RT-PCR, qRT-PCR, and WISH (Fig. 3a–f). A dose-dependency assay using TRAM and Met-ENK was performed, which demonstrated that higher concentrations (100 μM Met-ENK, 16 μM TRAM, 10 μL per fish) exerted stronger inhibitory effects on the expression of *lhx1a* and resulted in greater reductions in the number of RPCAs than lower concentrations (50 μM Met-ENK, 10 μM TRAM, 10 μL per fish), as observed through RT-PCR, qRT-PCR, and WISH (Fig. 3a–f). To further investigate the role of PENK-A activation, we generated a transgenic line, *Tg(hsp70l:penka)*, with endogenous overexpression of *penka* following heat shock (Fig. 3g). A previous study has demonstrated that heat shock accelerates the kidney regeneration response[23]. Upon heat shock at 2, 4, and 6 dpi, we found significant increases in the expression of *lhx1a* and the number of RPCAs in injured WT kidneys, whereas no such increases were observed in injured *Tg(hsp70l:penka)* kidneys at 7 dpi, as observed through RT-PCR and WISH (Fig. 3g–i). These findings suggest that PTECs can finely regulate the number of neonatal nephrons by modulating *penka* expression.

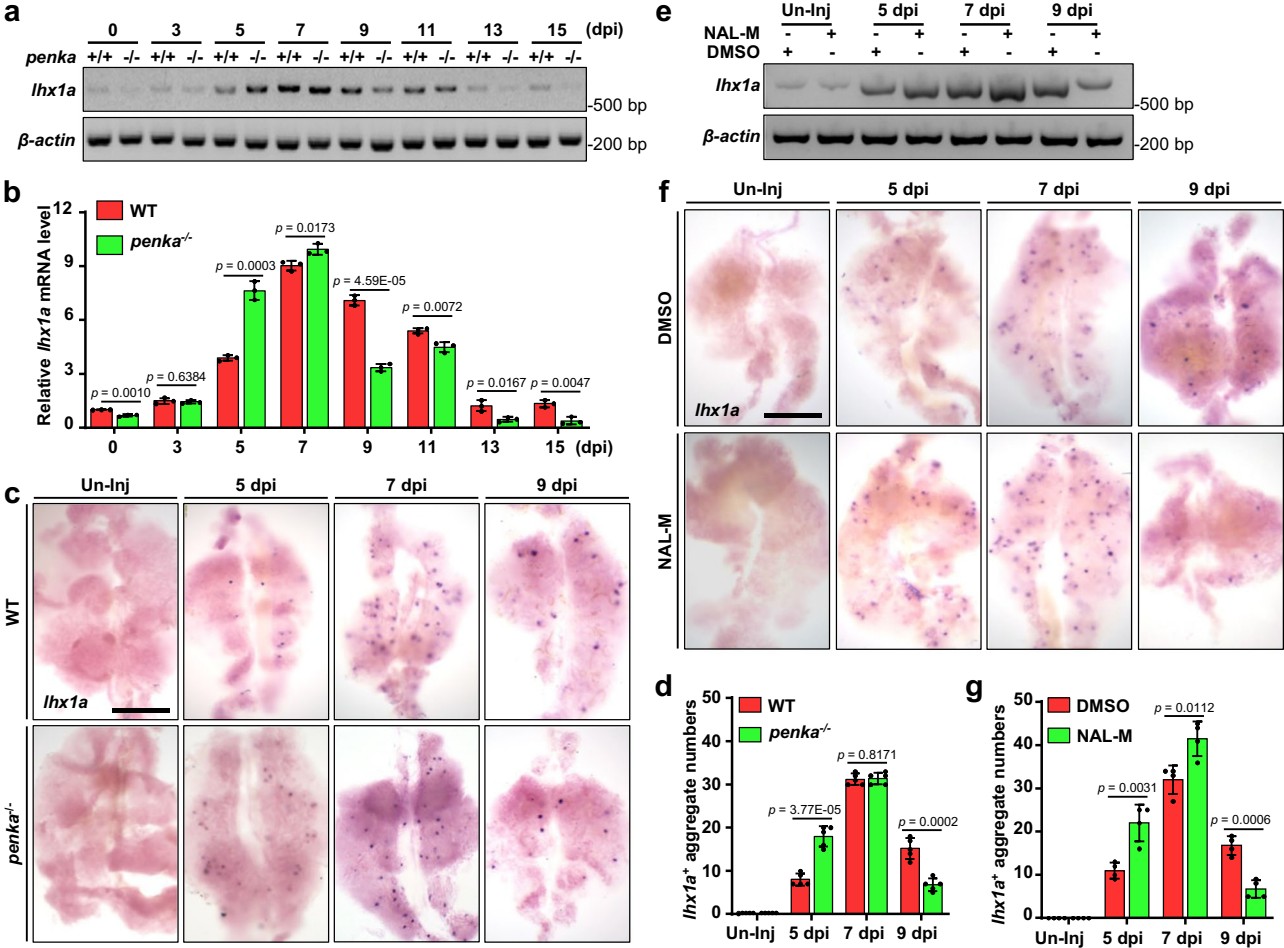

**Fig. 2 | PENK-A deficiency accelerates kidney regeneration. a** RT-PCR analysis of *lhx1a* expression in WT and *penka⁻/⁻* zebrafish kidneys during Gent-induced AKI. **b** The band intensities in a are normalized to that of the loading control, *β-actin*, and the relative expression levels of *lhx1a* were quantified (*n* = 3 biological replications per group). The data are presented as the fold change relative to the 0 dpi WT groups. **c** WISH analysis of *lhx1a* was performed in WT and *penka⁻/⁻* zebrafish kidneys without injury (Un-Inj) and at 5, 7, and 9 dpi. **d** Quantitation of *lhx1a⁺* RPCAs (blue points) per kidney (*n* = 5) for each condition in (**c**). **e**, **f** RT-PCR (**e**) and WISH (**f**) analyses of *lhx1a* in Un-Inj, 5, 7, and 9 dpi kidneys after administration of NAL-M or DMSO (*n* = 3 in **e**, *n* = 4 in **f**). **g** The *lhx1a⁺* RPCAs per kidney (*n* = 4) were quantified for each condition in (**f**). The data in (**b**), (**d**), and (**g**) were analyzed by two-sided t-test and are presented as mean values ± SD. Scale bars in (**c**) and (**f**), 600 μm. Source data are provided as a Source data file.

The expression of *penka* increased beginning on 9 dpi and returned to normal levels at 15 dpi, indicating its association with the termination stage of kidney regeneration. To investigate this relationship, we intraperitoneally injected TRAM (16 μM, 10 μL per fish) and Met-ENK (100 μM, 10 μL per fish) at 4 and 6 dpi, which correspond to the peak period of kidney regeneration. Consequently, the expression of *lhx1a* and the number of RPCAs significantly decreased, as observed through RT-PCR and WISH (Fig. 3j–m). To further support our findings, we employed a recently developed transgenic line, *Tg(lhx1a:DsRed)*, in which individual RPCs and RPCAs are labeled[25]. We examined the injured kidneys of adult *Tg(lhx1a:DsRed)* fish after treatment with Met-ENK (100 μM, 10 μL per fish), and observed a decrease in the number of *lhx1a⁺* RPCAs and a significant increase in the number of individual RPCs at 5 dpi (Fig. 3n, o). This experiment confirmed that early activation of PENK-A can prematurely terminate kidney regeneration.

**PENK-A affects kidney regeneration by regulating H₂O₂ production**

To elucidate the mechanisms underlying PENK-A's regulation of kidney regeneration, we analyzed the expression of PENK-A receptors. Upon analyzing the scRNA-seq data of adult zebrafish kidneys[38], we observed low expression levels of genes encoding canonical PENK-A receptors, such as δ-opioid receptors and μ-

opioid receptors. However, genes encoding noncanonical opioid receptors in the opioid growth factor receptor (*ogfr*) family[51], including *ogfr*, *ogfrl1*, and *ogfrl2*, exhibited high expression levels (Fig. 4a). Notably, these genes were predominantly expressed in renal medullary cells, such as macrophages, neutrophils, T cells and neurons (Fig. 4a). In our previous studies, we established that ROS, particularly H₂O₂, serve as the initial signals for kidney regeneration[29]. H₂O₂ production primarily occurs in renal medullary cells. We observed a gradual increase in the expression of the H₂O₂ synthase gene, *duox1*, and the production of H₂O₂ labeled with pentafluorobenzenesulfonyl fluorescein (PBSF) in the injured adult kidneys, which peaked at 5 dpi. Inhibition of H₂O₂ production impairs the formation of new nephrons[29]. To further confirm the location of H₂O₂ production, we utilized injured adult kidneys from the *Tg(cdh17:DsRed)* transgenic line, in which renal tubules are specifically marked[25,52]. We performed PBSF staining and observed that after 3 dpi, the majority of H₂O₂ was generated in the renal medulla (Fig. 4b). Interestingly, previous studies have demonstrated that exogenous opioids can hinder tissue regeneration by suppressing ROS production in mice, as exemplified by the regeneration of the inguinal fat pad[53]. Thus, based on these findings, we hypothesized that endogenous enkephalin PENK-A may exert its effects through a similar mechanism.

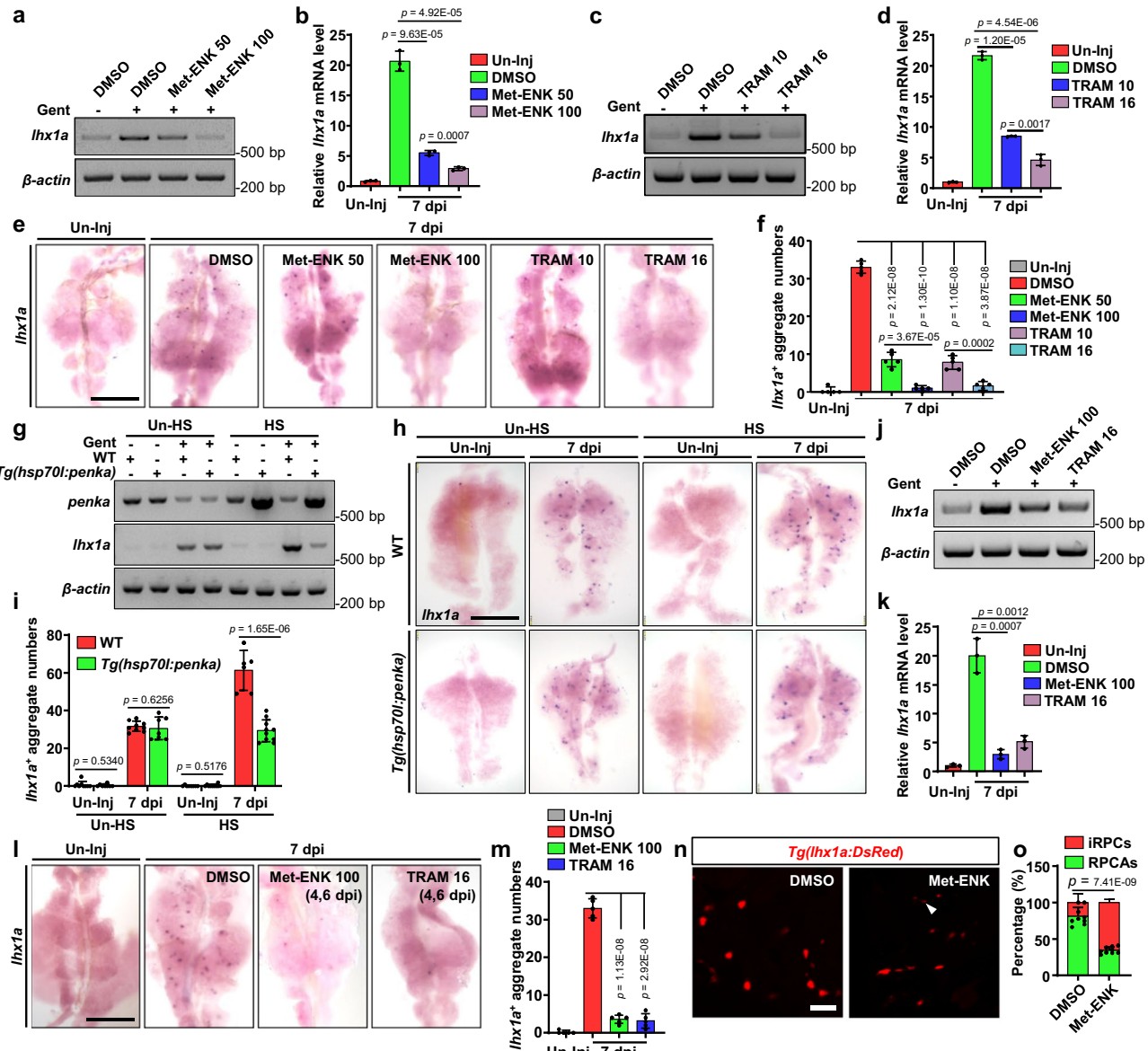

**Fig. 3 | PENK-A acts as a brake in kidney regeneration. a–e** RT-PCR (**a**, **c**), qRT-PCR (**b**, **d**), and WISH (**e**) analyses of *lhx1a* were performed on kidneys administered with Met-ENK (**a**, **b**) or TRAM (**c**, **d**) at various doses at 7 dpi after AKI (*n* = 3 in **b** and **d**). Met-ENK 50, 50 μM Met-ENK; Met-ENK 100, 100 μM Met-ENK; TRAM 10, 10 μM TRAM; and TRAM 16, 16 μM TRAM. **f** The *lhx1a*+ RPCAs per kidney (*n* = 5) were quantified for each condition in (**e**). The data in (**b**) and (**d**) are presented as the fold change relative to the Un-Inj group. **g** RT-PCR analysis of *penka* and *lhx1a* in WT and *Tg(hsp70l:penka)* zebrafish kidneys with heat shock (HS) or without HS (Un-HS) in the Un-Inj group or at 7 dpi (*n* = 3). **h** WISH analysis of *lhx1a* in WT and *Tg(hsp70l:penka)* zebrafish kidneys with HS (*n* = 6 in WT group, *n* = 10 in *Tg(hsp70l:penka)* group) or Un-HS (*n* = 9 in WT group, *n* = 6 in *Tg(hsp70l:penka)* group) in the Un-Inj group or at 7 dpi. **i** Quantitation of *lhx1a*+ RPCAs per kidney for

each condition in (**h**). **j–l** RT-PCR (**j**), qRT-PCR (**k**), and WISH (**l**) analyses of *lhx1a* at 7 dpi after administration (at 4 and 6 dpi) of Met-ENK 100 (100 μM Met-ENK, 10 μL per fish) or TRAM 16 (16 μM TRAM, 10 μL per fish) following AKI. The data in (**k**) (*n* = 3) are presented as the fold change relative to the Un-Inj group. **m** The *lhx1a*+ RPCAs per kidney (*n* = 5) were quantified for each condition in (**l**). **n** Confocal images showing adult *Tg(lhx1a:DsRed)* kidneys at 5 dpi after administration (at 2 and 4 dpi) of Met-ENK or DMSO following AKI (*n* = 9). **o** Quantitation of the individual RPCs (iRPCs, arrowhead) and RPCAs in (**n**). Data in (**b**), (**d**), (**f**), (**i**), (**k**), (**m**), and (**o**) were analyzed by two-sided t-test and are presented as mean values ± SD. Scale bars in (**e**), (**h**), and (**l**), 600 μm; (**n**) 50 μm. Source data are provided as a Source data file.

To investigate whether PTECs regulate the production of $H_2O_2$ through *penka*, we utilized a fluorimetric hydrogen peroxide assay kit to accurately measure the $H_2O_2$ levels in the kidneys. We injected NAL-M and observed that the $H_2O_2$ levels in the treatment group were higher than those in the control group in all stages (Fig. 4c). In addition, we assessed the $H_2O_2$ levels in adult *penka*$^{-1+24}$ mutant kidneys. Interestingly, we observed significantly elevated $H_2O_2$ levels in the injured kidneys of homozygous *penka*$^{-1+24}$ mutants throughout all stages of regeneration (Fig. 4d), while the $H_2O_2$ levels in injured WT kidneys remained lower. The injured kidneys of the heterozygous

*penka*$^{-1+24}$ mutants exhibited intermediate $H_2O_2$ levels (Fig. 4d). These results indicate a correlation between the *penka* expression level and $H_2O_2$ production.

To investigate the impact of PENK-A pathway reactivation on $H_2O_2$ production during regeneration, we examined regenerating adult kidneys treated with TRAM and Met-ENK or kidneys of *Tg(hsp70l:penka)* fish subjected to heat shock in at 2, 4, and 6 dpi. The results revealed that the levels of $H_2O_2$ were lower than those in the control group at all stages (Fig. 4e–g). In addition, a higher concentration of the agonist led to more significant inhibition of $H_2O_2$

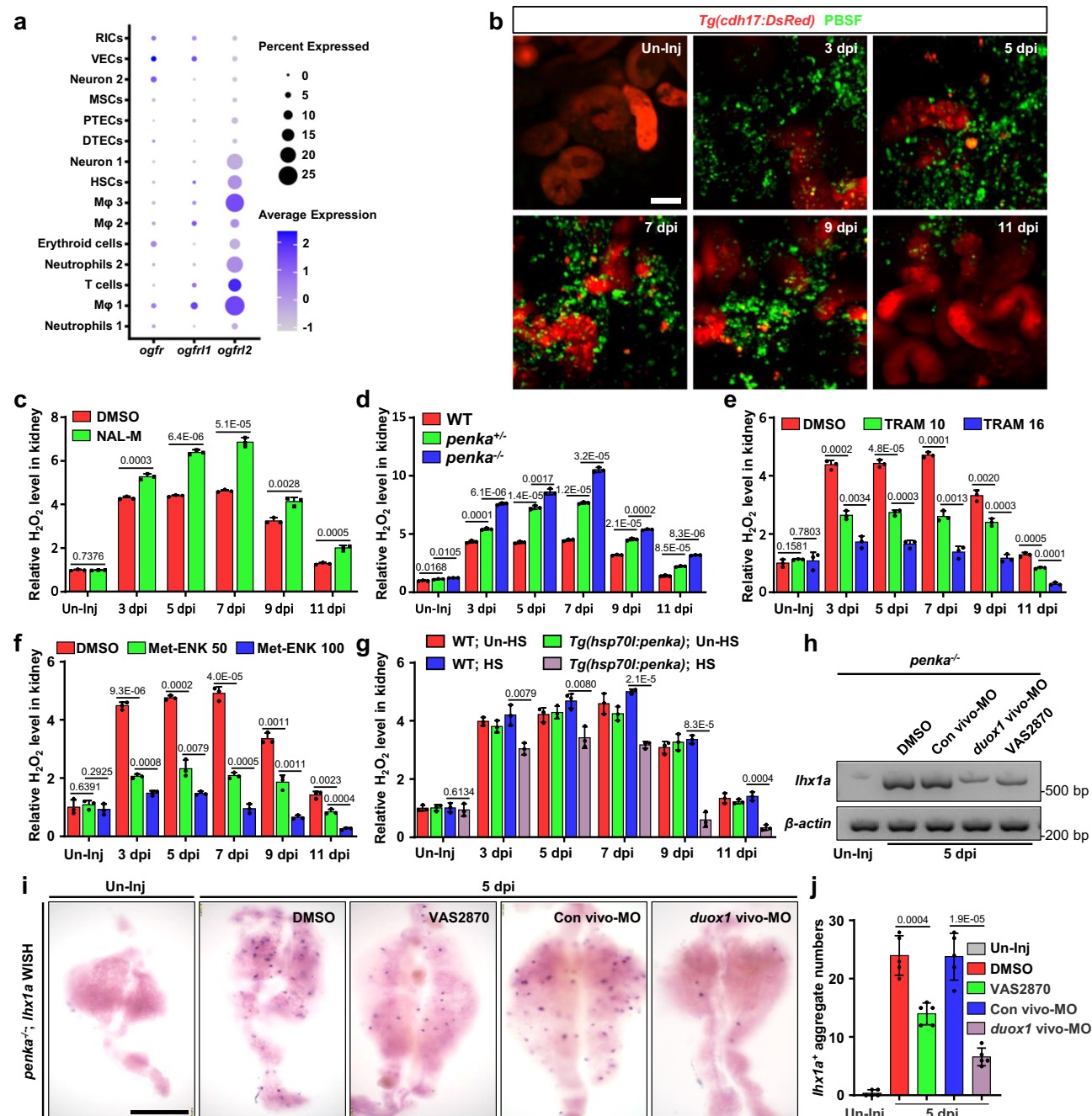

**Fig. 4 | PENK-A regulates H₂O₂ production. a** Gene expression of PENK-A receptors (*ogfr*, *ogfrl1*, and *ogfrl2*) in kidney cells. VECs (vascular endothelial cells), DTECs (distal tubular epithelial cells), Mφ (macrophages), HSCs (hematopoietic stem cells), MSCs (mucin-secreting cells), and RICs (renal interstitial cells). **b** Confocal images of H₂O₂ signal in adult *Tg(cdh17:DsRed)* kidneys after Gent-induced AKI, detected using the PBSF fluorescence probe. Scale bar, 50 μm. **c** Relative H₂O₂ concentration in kidneys after NAL-M or DMSO administration following AKI, presented as fold change relative to the DMSO-treated Un-Inj group (*n* = 3). **d** Relative H₂O₂ concentrations in the kidneys of WT, *penka⁺ᐟ⁻*, and *penka⁻ᐟ⁻* zebrafish following AKI, presented as fold change relative to the Un-Inj WT group (*n* = 3). **e** H₂O₂ concentration in kidneys after TRAM 10 (10 μM TRAM, 10 μL per fish), TRAM 16 (16 μM TRAM, 10 μL per fish), or DMSO administration following AKI, presented as fold change relative to the DMSO-treated Un-Inj groups (*n* = 3).

**f** H₂O₂ concentration in kidneys after Met-ENK 50 (50 μM Met-ENK, 10 μL per fish), Met-ENK 100 (100 μM Met-ENK, 10 μL per fish), or DMSO administration following AKI, presented as fold change relative to the DMSO-treated Un-Inj groups (*n* = 3). **g** H₂O₂ concentrations in the kidneys of WT and *Tg(hsp70l:penka)* with HS or without HS (Un-HS) following AKI, presented as fold change relative to the Un-HS and Un-Inj WT groups (*n* = 3). No significant differences between Un-HS WT, HS WT, and Un-HS *Tg(hsp70l:penka)* zebrafish were found using two-tailed t-test. **h, i** RT-PCR (**h**) and WISH (**i**) analyses of *lhx1a* in 5 dpi *penka⁻ᐟ⁻* kidneys after administration (at 2 and 4 dpi) of VAS2870, duox1 vivo-MO, Con vivo-MO, or DMSO after AKI, Scale bar, 600 μm. **j** Quantification of *lhx1a⁺* RPCAs per kidney (*n* = 5) for each condition in (**i**). Data in (**c**), (**d**), (**e**), (**f**), (**g**), and (**i**) were analyzed by two-sided t-test and are presented as mean values ± SD. *p* values are listed. Source data are provided as a Source data file.

production (Fig. 4e, f), demonstrating a dose-dependent relationship between PENK-A and H₂O₂. Our previous studies have demonstrated that injection of the NADPH oxidase antagonist VAS2870 or the H₂O₂ synthase *duox1* vivo-MO effectively inhibits H₂O₂ production in injured kidneys[29]. To confirm that *penka* deficiency accelerates kidney regeneration through H₂O₂, we intraperitoneally injected VAS2870 or *duox1* vivo-MO into *penka⁻⁺[24]* mutants. The rapid regeneration observed at 5 dpi in the mutants was inhibited (Fig. 4h–j), providing evidence that

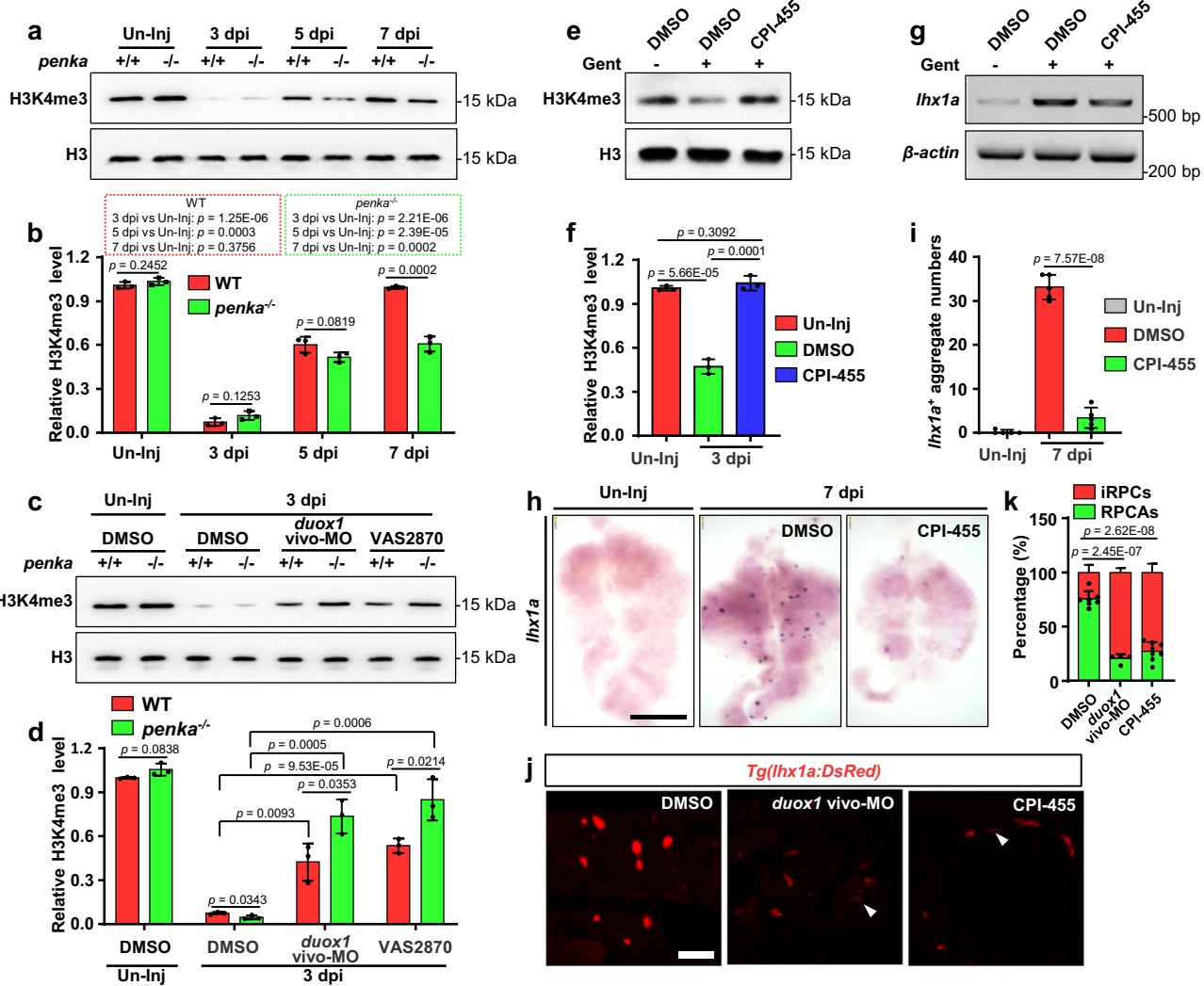

**Fig. 5 | The PENK-A–H₂O₂ pathway affects the remodeling of global H3K4me3 in kidney cells.** **a** Western blot analysis of H3K4me3 levels in WT and *penka⁻/⁻* zebrafish kidneys following AKI. **b** The protein band intensities in a were normalized to the loading control, Histone 3 (H3), and the relative expression levels of H3K4me3 were quantified (*n* = 3 biological replications per group). The data are presented as the fold change relative to the Un-Inj WT group. **c** Western blot analysis of H3K4me3 levels in WT and *penka⁻/⁻* zebrafish kidneys at 3 dpi after administration (at 2 dpi) of *duox1* vivo-MO, VAS2870, or DMSO following AKI. **d** The protein band intensities in (**c**) were normalized to the loading control, H3, and the relative expression levels of H3K4me3 were quantified (*n* = 3 biological replications per group). The data are presented as the fold change relative to the Un-Inj WT group. **e** Western blot analysis of H3K4me3 levels in the kidneys at 3 dpi after

administration (at 2 dpi) of CPI-455 or DMSO following AKI. **f** The protein band intensities in e were normalized to the loading control, H3, and the relative expression levels of H3K4me3 were quantified (*n* = 3). The data are presented as the fold change relative to the Un-Inj groups. **g**, **h** RT-PCR (**g**) and WISH (**h**) analyses of *lhx1a* at 7 dpi after administration (at 2, 4, and 6 dpi) of CPI-455 or DMSO following AKI (*n* = 3). **i** The *lhx1a⁺* RPCAs per kidney (*n* = 5) were quantified for each condition in (**h**). **j** Confocal images showing adult *Tg(lhx1a:DsRed)* kidneys at 5 dpi after administration (at 2 and 4 dpi) of *duox1* vivo-MO (*n* = 4), CPI-455 (*n* = 7), or DMSO (*n* = 8) following AKI. Scale bar, 100 μm. **k** Quantitation of individual RPCs (iRPCs, arrowheads) and RPCAs in (**j**). The data in (**b**), (**d**), (**f**), (**i**), and (**k**) were analyzed by two-sided t-test and are presented as mean values ± SD. Source data are provided as a Source data file.

PENK-A regulates kidney regeneration through H₂O₂. Overall, the findings indicate that during the initial stage of regeneration, decreased *penka* expression promotes H₂O₂ production, thereby accelerating kidney regeneration. However, in the late stage, as the number of neonatal nephrons increases, the expression of PENK-A also rises. This increase in PENK-A can inhibit the production of H₂O₂, thereby contributing to the termination of the kidney regeneration process.

### The PENK-A–H₂O₂ pathway regulates kidney regeneration by affecting the remodeling of global H3K4me3 in the kidneys

H₂O₂ is a relatively stable ROS, that can freely diffuse among cells[54]. Previous studies have indicated that H₂O₂ can induce downregulation of H3K4me3, that occurs at the promoter region and is associated with the

activation of nearby gene expression, in *Caenorhabditis elegans* and HeLa cells[55]. This suggests that H₂O₂ may play a role in gene switching and regulation of gene expression through histone modification. In light of this, we sought to investigate whether H₂O₂-induced changes have similar effects in the zebrafish kidneys. To address this, we investigated the changes in H3K4me3 levels during kidney regeneration in WT and *penka⁻¹⁺²⁴* mutant kidneys at 3, 5, and 7 dpi. Remarkably, among WT and *penka⁻¹⁺²⁴* mutant kidneys, our Western blot analyses revealed significantly lower H3K4me3 levels at 3 dpi in injured kidneys than in uninjured kidneys (Fig. 5a, b). Furthermore, in WT kidneys, a subsequent increase in H3K4me3 levels was observed at 5 dpi, and by 7 dpi, the difference from the levels in the uninjured group was not statistically significant (Fig. 5a, b). However, at 5 and 7 dpi, the H3K4me3 levels in *penka⁻¹⁺²⁴* mutant kidneys were lower than those in WT kidneys

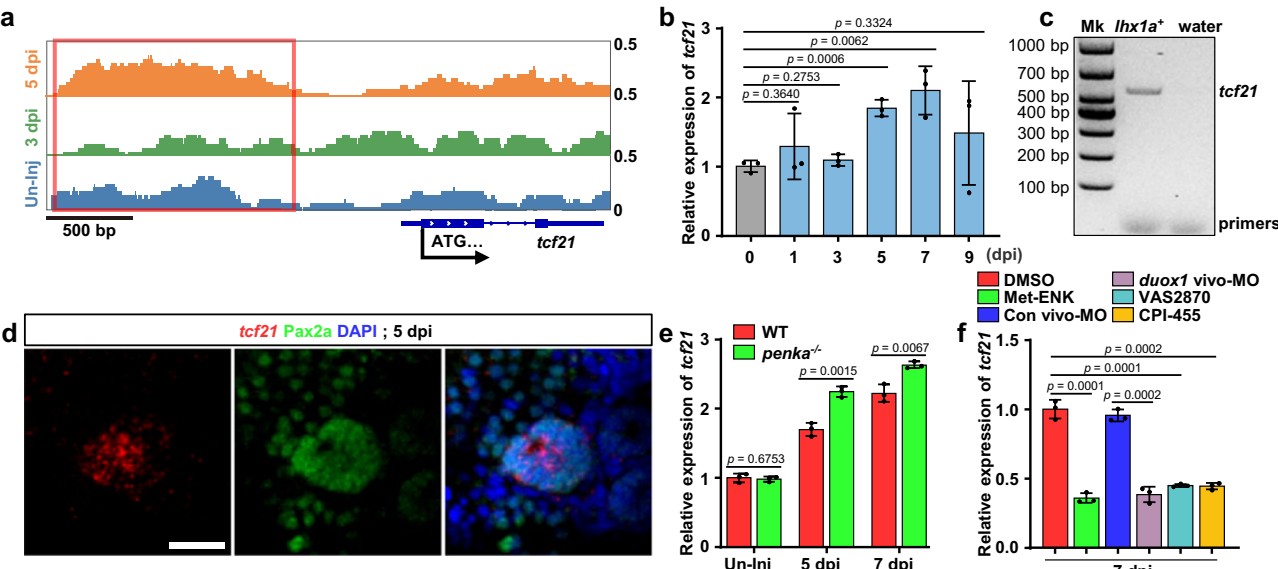

**Fig. 6 | The PENK-A–H$_2$O$_2$ pathway regulates *tcf21* expression through promoter H3K4me3 remodeling. a** ChIP-seq analysis of the H3K4me3 pattern in the promoter region of *tcf21*. The H3K4me3 level upstream of the ATG start codon (red box) was decreased significantly at 3 dpi and increased at 5 dpi. **b** qRT-PCR analysis of *tcf21* in zebrafish kidneys during AKI. The data were presented as the fold change relative to the 0 dpi group ($n = 3$ biological replications per group). **c** FACS coupled with RT-PCR analysis of *tcf21* expression in *lhx1a:DsRed*-labeled RPCs at 5 dpi after AKI. Water was used as the RT-PCR negative control. Mk, Marker. **d** Confocal images revealed that the combination of *tcf21* FISH with Pax2a immunofluorescence showed high expression of *tcf21* in Pax2a$^+$ RPCAs at 5 dpi after AKI. Scale bar, 50 μm. **e** qRT-PCR analysis of *tcf21* in WT and *penka$^{-/-}$* kidneys following AKI ($n = 3$ biological replications per group). The data were presented as the fold change relative to the 0 dpi WT group. **f** qRT-PCR analysis of *tcf21* in 7 dpi WT kidneys after administration (at 2, 4, and 6 dpi) of Met-ENK, Con vivo-MO, *duox1* vivo-MO, VAS2870, CPI-455, and DMSO following AKI ($n = 3$ biological replications per group). The data were presented as the fold change relative to the 7 dpi DMSO-treated group. The data in (**b**), (**e**), and (**f**) were analyzed by two-sided t-test and are presented as mean values ± SD. Source data are provided as a Source data file.

(Fig. 5a, b). These findings align with the temporal pattern of H$_2$O$_2$ generation and persistence. Consequently, we propose that the burst of H$_2$O$_2$ may induce remodeling of H3K4me3 in renal cells.

To investigate the relationship between H$_2$O$_2$ and H3K4me3 in the injured kidney, we examined the changes in H3K4me3 levels in samples where H$_2$O$_2$ production was inhibited. Inhibition of H$_2$O$_2$ production using VAS2870 or *duox1* vivo-MO in WT and *penka$^{-1+24}$* mutant kidneys prevented the decrease in H3K4me3 levels observed at 3 dpi in Western blot analyses (Fig. 5c, d). These findings indicate that H$_2$O$_2$ generation is necessary for the remodeling of H3K4me3. We also injected the H3K4me3 demethylase KDM5 inhibitor CPI-455[56] (80 μM, 10 μL per fish) intraperitoneally in WT zebrafish and found that the level of H3K4me3 in the treatment group was significantly higher than that in the control group at 3 dpi (Fig. 5e, f). In addition, the number of RPCAs was significantly lower in the treatment group compared to the control group at 7 dpi (Fig. 5g–i). These results suggest that the initial burst of H$_2$O$_2$ at 3 dpi triggers remodeling of H3K4me3 in kidney cells, which is crucial for kidney regeneration. In addition, they imply that *penka* regulates the remodeling of H3K4me3 by manipulating the production of H$_2$O$_2$.

### Remodeling of H3K4me3 promotes RPCA formation

During kidney regeneration, H$_2$O$_2$ is generated in cells localized in the renal medulla, where *lhx1a$^+$* RPCs are also found[22,29]. The aforementioned data reveal an increase in the number of individual RPCs and a decrease in the number of RPCAs in PENK-A agonist-treated regenerating kidneys (Fig. 3n, o). Therefore, we hypothesized that inhibiting H$_2$O$_2$ would yield similar effects. Upon knockdown of the H$_2$O$_2$ synthase *duox1* using *duox1* vivo-MO at 2 and 4 dpi in *Tg(lhx1a:DsRed)* kidneys, we observed a significantly lower number of *lhx1a$^+$* RPCAs in the treated group than in the control group at 5 dpi, while the number of individual *lhx1a$^+$* RPCs was higher in the treated group (Fig. 5j, k). These findings suggest that H$_2$O$_2$ influences the process of RPCs

aggregation to form RPCAs. In addition, we examined the effect of CPI-455 on RPCA formation and found that CPI-455 injection produced a phenotype similar to that elicited by *duox1* vivo-MO (Fig. 5j, k). These data indicate that H$_2$O$_2$ affects the aggregation of RPCs to generate RPCAs. RPCs undergo mesenchymal–epithelial transition (MET) during the aggregation process and subsequently differentiate into mature nephrons[25,26]. We speculate that H$_2$O$_2$ may activate the expression of genes crucial for the RPC MET process by increasing H3K4me3 levels in the promoter regions of these genes, ultimately leading to the promotion of RPCA formation.

### The PENK-A–H$_2$O$_2$ pathway controls kidney regeneration by regulating *tcf21*

To identify direct targets of H3K4me3 remodeling during kidney regeneration, we performed chromatin immunoprecipitation sequencing (ChIP-seq) to examine changes in H3K4me3 levels in all adult kidney cells at 0, 3, and 5 dpi. ChIP-seq analysis revealed significant changes in the H3K4me3 levels of genes associated with apoptosis and the response to stimuli during kidney regeneration (Supplementary Fig. S4a–c). In addition, we observed significant alterations in the promoter region of *tcf21* (Fig. 6a), a gene known to play a crucial role in MET[57]. Specifically, the H3K4me3 levels upstream of the *tcf21* ATG start codon exhibited a significant decrease at 3 dpi followed by an increase at 5 dpi. Based on these findings, we hypothesize that *tcf21* may be closely linked to the regulatory effects of PENK-A on kidney regeneration.

TCF21 is a helix-loop-helix domain-containing transcription factor that participates in regulating cell differentiation and cell fate transformation through MET[57]. We investigated the expression of the *tcf21* gene at different time points during kidney regeneration using qRT-PCR and observed that *tcf21* expression began to increase at 5 dpi, which coincided with the formation of RPCAs (Fig. 6b). To confirm whether *tcf21* is expressed in RPCs, we sorted *lhx1a$^+$* RPCs at 5 dpi using

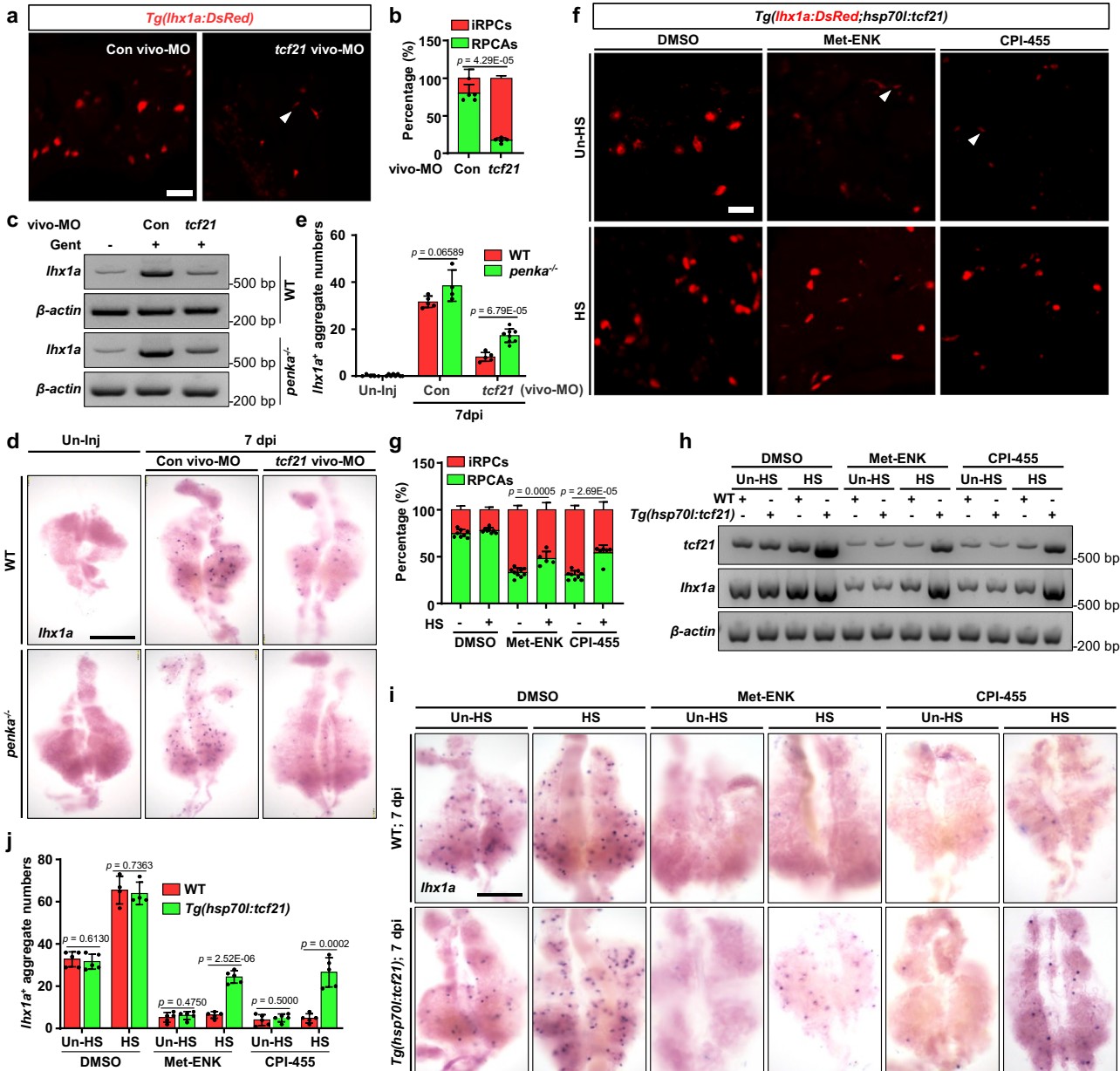

**Fig. 7 | The PENK-A–H₂O₂ pathway regulates kidney regeneration through** *tcf21.* **a** Confocal images showing adult *Tg(lhx1a:DsRed)* kidneys at 5 dpi afer administration (at 2 and 4 dpi) of *tcf21* vivo-MO, or Con vivo-MO following AKI (*n* = 5 biological replications per group). **b** Quantitation of individual RPCs (iRPCs, arrowhead) and RPCAs in a. **c, d** RT-PCR (*n* = 3) (**c**) and WISH (**d**) analyses of *lhx1a* in 7 dpi WT and *penka⁻ᐟ⁻* kidneys with administration (at 2, 4, and 6 dpi) of *tcf21* vivo-MO (*n* = 5 in WT group, *n* = 8 in *penka⁻ᐟ⁻* group) or Con vivo-MO ((*n* = 5 in WT group, n = 4 in *penka⁻ᐟ⁻* group) following AKI. **e** The *lhx1a⁺* RPCAs per kidney were quantified for each condition in d. **f** Confocal images showing 5 dpi *Tg(lhx1a:Ds-Red;hsp70l:tcf21)* kidneys with HS (at 2 and 4 dpi) or Un-HS after administration (at 2 and 4 dpi) of Met-ENK (*n* = 9 in Un-HS group, *n* = 5 in HS group), CPI-455 (*n* = 9 in Un-

HS group, *n* = 6 in HS group), or DMSO (*n* = 8) following AKI. **g** Quantitation of individual RPCs (iRPCs, arrowheads) and RPCAs in (**f**). **h** RT-PCR analysis of *tcf21* and *lhx1a* in 7 dpi WT and *Tg(hsp70l:tcf21)* kidneys with HS (at 2, 4, and 6 dpi) or Un-HS after administration (at 2, 4, and 6 dpi) of Met-ENK, CPI-455 or DMSO following AKI (*n* = 3). **i** WISH analysis of *lhx1a* in 7 dpi WT and *Tg(hsp70l:tcf21)* kidneys with HS (at 2 and 4 dpi) or Un-HS after administration (at 2, 4, and 6 dpi) of Met-ENK (*n* = 5), CPI-455 (*n* = 5) or DMSO (*n* = 5 in Un-HS group, *n* = 4 in HS group) following AKI. **j** Quantitation of *lhx1a⁺* RPCAs per kidney was performed for each condition in (**i**). Data in (**b**), (**e**), (**g**), and (**j**) were analyzed by two-sided t-test and are presented as mean values ± SD. Scale bars in (**a**) and (**f**), 50 μm; (**d**) and (**i**), 600 μm. Source data are provided as a Source data file.

fluorescence-activated cell sorting (FACS) and performed RT-PCR using *Tg(lhx1a:DsRed)*-injured kidneys. The results demonstrated high expression of *tcf21* in the sorted *lhx1a⁺* cells (Fig. 6c). In addition, we employed a combined approach with *tcf21* FISH and Pax2a (a marker for RPCAs[22,25,29]) immunofluorescence in injured kidneys and observed high expression of *tcf21* in Pax2a⁺ RPCAs (Fig. 6d). We also examined the expression of *tcf21* in *penka⁻¹⁺²⁴* mutant kidneys through qRT-PCR analysis. In comparison to WT, the *penka⁻¹⁺²⁴* mutant kidneys displayed an elevated *tcf21* expression at 5 and 7 dpi (Fig. 6e). However, in the

Met-ENK treated WT group, the qRT-PCR analysis results indicated a significant decrease in the expression of *tcf21* at 7 dpi (Fig. 6f). The intraperitoneal injection of VAS2870 or *duox1* vivo-MO into WT fish also resulted in the suppression of the upregulation of *tcf21* expression (Fig. 6f). Furthermore, in the CPI-455 treated group, the expression of *tcf21* did not show an increase when H3K4me3 remodeling was inhibited (Fig. 6f), which corresponded to the change in *lhx1a* expression (Fig. 5g). These findings suggest that PENK-A influences the formation of RPCAs through its effects on *tcf21*.

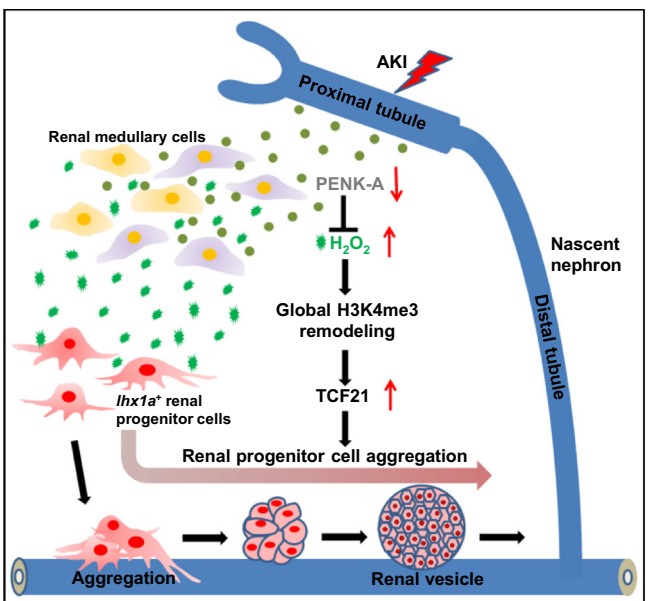

**Fig. 8 | Graphical abstract summarizing the research findings.** PTEC-expressed PENK-A levels decreased with the loss of PTs after AKI. This decrease triggered an increase in H$_2$O$_2$ production, resulting in the expansion of RPC aggregation. This process occurred by upregulating *tcf21* expression through remodeling of H3K4me3 in the *tcf21* promoter. In the later stage of regeneration, as PTECs recovered, elevated *penka* expression suppressed the production of H$_2$O$_2$, facilitating the termination of kidney regeneration.

Due to the inability of zebrafish *tcf21* mutants to survive to adulthood, we suppressed *tcf21* expression through intraperitoneal injection of *tcf21* vivo-MO[58] (25 μM, 10 μL per fish) (Supplementary Fig. S5a, b). Treatment with *tcf21* vivo-MO significantly reduced the number of RPCAs, while increasing the number of individual RPCs in *Tg(lhx1a:DsRed)* kidneys (Fig. 7a, b). WISH and RT-PCR analyses also revealed a significant decrease in the expression of *lhx1a* and a reduction in the number of RPCAs in the *tcf21* vivo-MO group compared to the control group (Fig. 7c–e). Moreover, the results of *lhx1a* RT-PCR and WISH in *penka*$^{-1+24}$ mutant kidneys injected with *tcf21* vivo-MO also indicated a reduction in RPCAs formation (Fig. 7c–e). In addition, we constructed a transgenic line, *Tg(hsp70l:tcf21)*, that can effectively overexpress *tcf21* after heat shock (Fig. 7h). Heat shock was conducted at 2 and 4 dpi, and we found significantly more RPCAs in the injured kidneys of the *Tg(hsp70l:tcf21;lhx1a:DsRed)* fish than in those of control fish (Fig. 7f). RT-PCR and WISH also revealed an increase in *lhx1a* expression and a significant increase in RPCAs (Fig. 7h–j). Furthermore, we observed that activating the PENK-A signaling pathway through intraperitoneal injection of Met-ENK (100 μM, 10 μL per fish) restored the number of RPCAs in the injured kidneys of heat-shocked *Tg(hsp70l:tcf21;lhx1a:DsRed)* fish (Fig. 7f). *lhx1a* RT-PCR and WISH confirmed these findings (Fig. 7h–j). When treated with CPI-455 (80 μM, 10 μL per fish) in heat-shocked *Tg(hsp70l:tcf21;lhx1a:DsRed)* fish, it was found that the overexpression of *tcf21* could rescue the inhibition of RPC aggregation by CPI-455 (Fig. 7f) and reactivate kidney regeneration (Fig. 7h–j). These results provide further evidence that PENK-A regulates kidney regeneration through its effects on *tcf21*.

In general, the results indicate that a sudden loss of PENK-A in PTECs can trigger a burst of H$_2$O$_2$ production within the kidney. The burst of H$_2$O$_2$ can modify the H3K4me3 levels at the *tcf21* promoter region, altering the activity of the *tcf21* promoter and promoting the expression of *tcf21*, thereby facilitating the expansion of RPCAs. During the later stages of regeneration, as PTECs recover, the increased expression of *penka* inhibits H$_2$O$_2$ production, thereby promoting the termination of kidney regeneration (Fig. 8). Thus, PTECs can regulate

their own regeneration through a negative feedback mechanism mediated by the PENK-A signaling pathway.

## Discussion

In this study, we investigated the involvement of endogenous opioids in zebrafish kidney regeneration using genetic approaches. Our findings demonstrate that *penka* expression is responsive to tissue damage during zebrafish kidney regeneration. A decrease in *penka* expression creates a conducive environment for the production of substantial amounts of H$_2$O$_2$ within the kidneys. Through analysis of a *penka* mutant, we observed increased H$_2$O$_2$ production in the kidneys of homozygous mutant adult zebrafish accompanied by early activation of kidney regeneration. The H$_2$O$_2$ levels in heterozygous mutants fell between those of WT and homozygous mutants, indicating a dependency on *penka* expression levels for H$_2$O$_2$ production. Given the combined results of pharmacological treatments and *penka* over-expression experiments, we conclude that PTECs can finely regulate H$_2$O$_2$ production by precisely controlling *penka* expression, thereby influencing kidney regeneration.

Clinical studies have provided evidence of a positive correlation between increased PENK and exacerbation of AKI. Higher concentrations of PENK may hinder AKI repair process[32,33,35]. In zebrafish, we observed a significant decrease in PENK-A expression prior to kidney tissue regeneration, suggesting that PENK-A could be an essential molecular factor in achieving rapid tissue repair following kidney injury in zebrafish. The role of ROS in the pathogenesis of kidney diseases has been well-documented. Elevated ROS levels contribute to renal inflammation, renal tubular injury, and renal fibrosis[30,31]. Consequently, screening for drugs that can inhibit excessive ROS production holds promise for the treatment of kidney diseases. Our study reveals the significant regulatory role of endogenous PENK-A in renal H$_2$O$_2$ production, positioning PENK as a potential drug target for kidney disease treatment. Currently, perioperative pain relief protocols routinely involve the use of opioid analogs as exogenous agents[59]. Further investigation of the relationship between PENK and the development and progression of kidney disease could enable these drugs to be utilized in the clinical management of kidney disease.

In a previous study on an AKI model induced by lipopolysaccharide (LPS), the overall level of H3K4me3 was observed to be significantly lower than that in compared to the control group[60]. However, the underlying reason for this decline remains unclear. Our investigation revealed a direct relationship between the H$_2$O$_2$ burst at 3 dpi and the decrease in H3K4me3 level at 3 dpi in the zebrafish kidneys. Blocking H$_2$O$_2$ production effectively inhibited the alterations in H3K4me3 levels. Notably, we observed that the levels of H3K4me3 began to rise at 5 and 7 dpi, indicating that the abrupt change in H$_2$O$_2$ production profoundly induced H3K4me3 remodeling in the kidney. Previous studies have also demonstrated that transient ROS exposure can modulate the global levels of H3K4me3 in both *Caenorhabditis elegans* and mammalian cells, leading to enhanced stress resistance[55]. Consequently, we propose that H3K4me3 remodeling can reshape the gene expression pattern of kidney cells and drive their participation in the process of kidney regeneration. By screening genes associated with alterations in renal H3K4me3 levels, we identified *tcf21* as a direct influencer of the regeneration process in RPCs. Knockdown of *tcf21* expression effectively hindered the formation of RPCAs. However, we acknowledge that *tcf21* may not be the sole factor influenced by H3K4me3 remodeling, and further investigations are warranted to elucidate the involved signaling pathways.

In this study, we made the significant discovery that PENK-A expressed by PTECs plays a pivotal role in determining the pace of kidney regeneration through its control over H$_2$O$_2$ production. A burst in H$_2$O$_2$ triggers remodeling of H3K4me3 in kidney cells. We further demonstrated that this remodeling affects the expression of *tcf21* in RPCs, which is a critical gene involved in the formation of RPCAs.

Collectively, our findings unveil the inhibitory role of PENK-A in kidney regeneration. Leveraging these insights will enable the design of safe and effective clinical interventions for the treatment of renal diseases.

## Methods

### Zebrafish husbandry and lines used

Zebrafish were produced, grown, and maintained according to standard protocols[61]. The following transgenic lines were used in this study: *Tg(lhx1a:DsRed)*[25], *Tg(cdh17:DsRed)*[25,52], *Tg(gtshβ:GFP)*[45], *Tg(hsp70l:penka)*, and *Tg(hsp70l:tcf21)*. Adult zebrafish aged between 3 to 12 months were used for the experiments, and approximately equal sex ratios were employed. Anesthesia was administered using 0.0168% buffered tricaine methanesulfonate (MS-222, Sigma). The AB strain of zebrafish served as the WT control for this study. Animal care and use protocol was approved by the Institutional Animal Care and Use Committee of the Army Medical University, China (SYXK-PLA-2007035).

### Single-cell gene expression profiling

For analysis of the expression of *penka* and its receptors in zebrafish kidneys, we utilized a portion of recently published scRNA-seq data (GSE100910)[38]. The Seurat package (version 4.2.0) was employed to perform data normalization, dimensionality reduction, clustering, and differential expression analysis.

### Human renal biopsy samples

Renal biopsies were performed as part of routine clinical diagnostic investigations. Three patients with AKI and three patients with no detectable lesions verified by renal biopsy were enrolled in this study from the Department of Nephrology, Xinqiao Hospital, Chongqing, China. Patients with inflammatory and autoimmune-associated diseases, diabetes, polycystic kidney disease, and pregnancy were excluded from the study. Kidney biopsies were obtained from these patients for FISH analysis. The human studies conducted in this research were approved by the Ethics Committee of Xinqiao Hospital, Army Medical University (No. 2023-YAN-143-01). The study design and conduct adhered to all applicable regulations concerning the use of human participants and were in accordance with the principles set forth in the Declaration of Helsinki. Informed written consent was provided by all participants. Furthermore, the study complies with the guidelines provided by the Ministry of Science and Technology (MOST) for the Review and Approval of Human Genetic Resources.

### FISH of human renal biopsy samples

The paraffin-embedded renal biopsy samples were sectioned to a thickness of 3 μm. These sections were permeabilized with proteinase K (10 μg/mL, Roche) in PBS with 0.1% Tween-20 (PBT) for 10 min with gentle rocking. Digoxigenin-labeled riboprobes were synthesized from cDNA fragments containing the sequence of human *PENK* (Supplementary Table S1). To detect the probes, we utilized an anti-digoxigenin-peroxidase antibody (Roche, 11207733910) and a TSA Plus Fluorescein system (PerkinElmer, NEL741001KT). Following FISH, the sections were stained with LTL (10 μg/mL, Vector Laboratories, FL −1321-2). Finally, images were captured using a Nikon A1 confocal microscope. Detailed antibody information is listed in Supplementary Table S2.

### Zebrafish AKI model

Intraperitoneal injection of Gent was employed to induce AKI in adult zebrafish following a previously described method[25]. In brief, Gent (2.7 μg/μL, 20 μL per fish), diluted in water, was intraperitoneally injected into the WT line or other zebrafish lines. Each injected zebrafish was then placed into an individual container. Zebrafish exhibiting proteinuria at 1 dpi were selected for subsequent experiments.

### Nephron counting

Each nephron contained only one segment of *gtshβ:GFP*-labeled PTs[45]. Therefore, the number of *gtshβ:GFP*-labeled PTs segment was used to counting the number of nephrons. To perform the counting, kidneys were carefully removed from the adult *Tg(gtshβ:GFP)* fish and imaged in a 1.5 mm by 1.5 mm area using a Nikon A1 confocal microscope. The total number of GFP positive segments was determined using ImageJ. Before counting, all the pictures were relabeled, and the person conducting the analysis had no knowledge of the treatment conditions for each sample. This step was taken to minimize subjective bias and maintain objectivity and accuracy in the analysis. To assess the variation in the number of nephrons following Gent-induced AKI, the baseline number of nephrons in the uninjured kidneys was considered 100%. Subsequently, the percentage change in nephrons after AKI compared to the uninjured condition was utilized to depict the extent of kidney damage.

### CRISPR–Cas9 mutagenesis

The CRISPR/Cas9 target site in *penka* was selected using the CRISPR design tool[46] (https://www.crisprscan.org/). The guide RNA targeted the sequence 5′-GGCTTCATGAAGCGTTACGGCGG-3′ (target underlined) in *penka* exon 2. To construct the gRNA, the T7 promoter and gRNA scaffold were cloned into the pMD 19-T vector. The DNA template for gRNA synthesis was obtained by PCR amplification. Subsequently, the gRNA was synthesized using a Hiscribe T7 High Yield RNA Synthesis Kit (NEB, E2040S). For CRISPR/Cas9 mutagenesis, a mixed solution containing 1 nL of Cas9 protein (0.7 μM, NEB, M0646T) and gRNA (75 ng/μL) was injected into one-cell stage WT embryos. After the injection, genomic DNA of the embryos was obtained at 24 h post-fertilization (hpf) to check for mutagenesis at the target site. The primers used to amplify the *penka* gene are listed in Supplementary Table S1. The PCR products were then subjected to Sanger sequencing to confirm mutagenesis. Once the mutagenesis was confirmed, other injected embryos were raised to adulthood as F0 fish. The F1 generation were obtained by mating F0 fish with WT zebrafish and raising them to adulthood. Genomic DNA was extracted from tail tissue of the F1 fish and analyzed using methods mentioned above. Two *penka* mutant alleles (*penka*[−1+24] and *penka*[−8]) were identified through PCR and DNA sequencing. These alleles were subsequently chosen for further propagation, leading to the establishment of stable *penka* mutant lines to facilitate subsequent analyses. To assess off-target effects, CRISPRScan was used to predict potential off-target sites[46,47], and the top five sites were selected for sequencing analysis. Primers and sequencing data were listed in Source data file.

### WISH

WISH was conducted following previously described methods[25,29]. Briefly, after removing the internal organs, except for the kidneys, zebrafish were fixed overnight with 4% paraformaldehyde (PFA). After three washes with PBT, the fixed kidneys were dissected from the body and permeabilized with proteinase K (10 μg/mL, Roche, 3508838) in PBT with rocking for 1 h. Digoxigenin-labeled riboprobes were generated from cDNA fragments containing the sequences of zebrafish *lhx1a*[25,29]. An anti-digoxigenin-alkaline phosphatase antibody (Roche, 11093274910) and NBT/BCIP substrate (Roche, 11681451001) were used to detect the probe. Images were captured using a BX3-CBH microscope. Detailed antibody information is listed in Supplementary Table S2.

### Combined FISH and immunofluorescence in zebrafish kidneys

Combined FISH and immunofluorescence were performed following previously described methods[25,62]. Briefly, kidneys were harvested and fixed in 4% PFA overnight at 4 °C, and then frozen sections were created at 100-μm thickness. The sections were permeabilized with proteinase K (10 μg/mL, Roche) in PBT for 20 min with rocking.

Digoxigenin-labeled riboprobes were generated from cDNA fragments comprising the sequences of zebrafish *tcf21*, *slc20a1a*, *trpm7*, and *penka* (Supplementary Table S1). An anti-digoxigenin-peroxidase antibody (Roche, 11207733910) and a TSA Plus Fluorescein system (PerkinElmer, NEL741001KT) were used to detect the probes. After FISH, the sections were stained for immunofluorescence. The primary antibodies used were anti-Met-ENK (Abcam, ab22620) and anti-Pax2a (Abcam, ab229318). The secondary antibody goat anti-rabbit IgG H&L Alexa Fluor 633 (Invitrogen, A21070) was used at a dilution of 1:500. Images were captured using a Nikon A1 confocal microscope. Detailed antibody information is listed in Supplementary Table S2.

### RT-PCR and qRT-PCR
RNA was extracted from kidney tissues using TRIzol reagent (Invitrogen, 15596018). A Prime Script II 1st strand cDNA Synthesis Kit (Takara, 9767) was used to synthesize cDNA, which was then subjected to PCR using Taq Master Mix (Vazyme, p112-01) for RT-PCR or TB Green Premix EX Taq II (Takara, RR820A) for qRT-PCR. All primers for *lhx1a*[25], *penka*, *tcf21*, and *β-actin*[25] are listed in Supplementary Table S1. Gene expression was normalized to *β-actin* mRNA expression. Full scan blots are provided as a Source data file.

### Western blotting
Zebrafish kidneys were obtained and homogenized with a 1 mL syringe and needle in cell lysis buffer (50 mM PBS, pH 7.4, 1% SDS, and 0.5% Triton X-100) containing protease inhibitor (Beyotime, P1005). The lysate was centrifuged at $12,000 \times g$ for 20 min at 4 °C, and the resulting supernatant was used as the protein sample. The protein concentration was determined using a bicinchoninic acid protein assay kit (CWBIO, CW0014S). Western blotting was carried out following standard protocols. The levels of total H3K4me3 and histone H3 were detected using the enhanced chemiluminescence (ECL) method with the antibodies listed in Supplementary Table S2. Full scan blots are provided as a Source data file.

### Pharmaceutical treatment
During kidney regeneration, TRAM (10 or 16 μM, 10 μL per fish), NAL-M (2.0 μM, 10 μL per fish), Met-ENK (50 or 100 μM, 10 μL per fish), VAS2870 (8 μM, 10 μL per fish), CPI-455 (80 μM, 10 μL per fish), or 0.1% DMSO (10 μL per fish) was intraperitoneally injected individually every other day starting from 2 dpi until the kidneys were collected for further analysis. To test the termination signal of kidney regeneration, TRAM, Met-ENK, or 0.1% DMSO (10 μL per fish) was intraperitoneally injected into zebrafish individually at 4 and 6 dpi. Kidneys were collected at 7 dpi for subsequent experiments. Detailed information about these inhibitors is listed in Supplementary Table S2.

### Vivo-MO
The *duox1* vivo-MO[29], *tcf21* vivo-MO[58], *penka* vivo-MO, and control vivo-MO[29] were designed as previously described and are listed in Supplementary Table S1. To verify the efficiency of *tcf21* vivo-MO and *penka* vivo-MO (two splice blocking morpholinos) in adult kidneys, *tcf21* vivo-MO (25 μM, 10 μL per fish) or *penka* vivo-MO (100 μM, 10 μL per fish) was intraperitoneally injected into zebrafish at 2, 4, and 6 dpi, and kidneys were collected at 7 dpi for RNA extraction and RT-PCR using the identification primers (Supplementary Table S1). During kidney regeneration, the vivo-MOs were intraperitoneally injected every other day beginning on 2 dpi until the kidneys were collected for further analysis.

### Generation of transgenic zebrafish lines
The coding sequence of *penka* or *tcf21* was amplified using PCR. The resulting fragments were then directionally cloned into the SalI/NotI site of the *hsp70l-loxP-mCherry-STOP-loxP-H2B-GFP_cryaa-cerulean/pI-*

*SceI* plasmid, leading to the generation of the *hsp70l:penka_cryaa-cerulean/pI-SceI* or *hsp70l:tcf21_cryaa-cerulean/pI-SceI* plasmid. For the transgenesis process, 1 nL of the injection mix containing 30 pg of each plasmid DNA and I-SceI restriction enzyme (2 U/μL, NEB, R0694S) was injected into one-cell-stage embryos. At ~96 h post-fertilization, we employed fluorescence microscopy to screen embryos displaying robust expression of blue fluorescent protein (cerulean) in their eyes. These embryos were then raised to generate F0 transgenic fish. The F1 generation was obtained by mating F0 transgenic fish with WT zebrafish. Based on whether they expressed the blue fluorescent protein cerulean in their eyes, four F1 fish were identified for each transgenic line. Subsequently, these F1 fish were crossed with WT zebrafish, and the resulting offspring were subjected to heat-shock on the third day after birth using the method described below. Total RNA was extracted from these heat-shocked embryos and converted into cDNA. Subsequently, the expression levels of *penka* or *tcf21* were analyzed using RT-PCR in these heat-shocked embryos. The fish line with the highest expression level of *penka* or *tcf21* was selected for further propagation, establishing the stable *Tg(hsp70l:penka)* or *Tg(hsp70l:tcf21)* lines for further analyses.

### Heat shock treatment
For heat shock treatment, zebrafish were transferred to preheated system water at 39 °C and kept at 39 °C for 30 min. After heat shock, the zebrafish were returned to the 28.5 °C system water. Zebrafish were heat shocked every other day after 2 dpi until the kidneys were collected for subsequent experiments.

### Imaging and quantification of *lhx1a*+ individual RPCs and RPCAs
Kidneys were carefully removed from the adult *Tg(lhx1a:DsRed)* zebrafish and imaged using a Nikon A1 confocal microscope. The individual DsRed+ cells and DsRed+ RPCAs were counted using ImageJ. Before counting, all the pictures were relabeled to ensure that the analyst conducting the analysis remained blind to the treatment conditions of each sample. This precautionary measure was implemented to minimize subjective bias and maintain objectivity and accuracy during the analysis.

### in situ $H_2O_2$ imaging
For in situ $H_2O_2$ imaging, adult *Tg(cdh17:DsRed)* zebrafish were intraperitoneally injected with a PBSF solution (Santa Cruz, SC-205429A; 100 μM, 10 μL per fish) to detect the location and level of $H_2O_2$. Three hours post-injection, the fish were anesthetized with 0.016% MS-222, sacrificed, and dissected to obtain the kidneys. The kidneys were then photographed using a Nikon A1 confocal microscope.

### $H_2O_2$ level measurement
$H_2O_2$ levels in adult zebrafish kidneys were assessed using a fluorimetric hydrogen peroxide assay kit (Sigma, MAK166) following the manufacturer's instructions. Briefly, kidney protein lysates were diluted to 1 mg/mL in NP-40 lysis buffer without protease inhibitors to avoid interference with the assay. In 96-well plates, 50 μL of each sample and standards of 0, 0.1, 0.3, 1, 3, and 10 μM $H_2O_2$ were added. Each reaction was then mixed with 50 μL of the reaction mixture, which consisted of assay buffer with 20 units/mL horseradish peroxidase and 1% infrared fluorometric peroxidase substrate. The plates were incubated at room temperature for 10 min and subsequently read fluorometrically at 640 nm excitation and 680 nm emission using a SpectraMAX M3 (Molecular Devices).

### FACS
To obtain *lhx1a:DsRed*-labeled cells, kidneys from ten *Tg(lhx1a:DsRed)* zebrafish were manually dissected in PBS and 0.05% trypsin-EDTA solution at 5 dpi. DsRed positive cells were sorted using a MoFlo XDP flow cytometer (Beckman) and collected for RNA extraction.

## ChIP-seq

Total DNA from uninjured zebrafish kidneys and injured zebrafish kidneys at 3 and 5 dpi was used for ChIP assays. For each assay, ten kidneys were utilized for total DNA extraction. The extracted DNA was then sheared into fragments ranging from 100 to 500 base pairs using an ultrasonic crusher. In the ChIP pull-down assay, 25 μg of chromatin samples (in equal volume) were incubated with 2.5 μg of specific antibodies (anti-H3K4me3 from CST, Cat# 9751) or rabbit IgG (as a negative control, from CST, Cat# 66326) for 24 h at 4 °C on a vertical rotor. The antibody–DNA–protein complexes were subsequently immunoprecipitated using a CUT&RUN Assay Kit (CST, Cat# 86652) following an agarose bead wash and incubation of the complex with the beads for 3 h at 4 °C. After coincubation, the beads were washed three times using incubation buffer. The DNA fragments were then extracted using the phenolchloroform method to construct the Illumina sequencing library. Library construction was performed by Novogene Corporation (Beijing, China). Subsequently, pair-end sequencing of the samples was conducted on the Illumina platform. The quality of the sequencing library was assessed using the Agilent Bioanalyzer 2100 system.

## ChIP-seq data analysis

The sequencing reads for ChIP and input DNA were aligned to the zebrafish reference genome files (GRCz11). The GRCz11 index was built using Burrows–Wheeler Aligner (BWA, v0.7.12), and clean reads were then mapped to the reference genome using BWA-MEM (v0.7.12). To build an enrichment model and predict fragment size, a specific number of windows were used as samples. Subsequently, peak calling analysis was performed based on the predicted fragment size. ChIPseeker was employed to retrieve the nearest genes surrounding the identified peaks and annotate the genomic regions associated with each peak. The ChIPseeker tool was utilized to confirm peak-related genes, and Gene Ontology (GO) enrichment analysis was conducted to identify enriched functions. In addition, KOBAS software was used to test the statistical enrichment of peak-related genes in Kyoto Encyclopedia of Genes and Genomes (KEGG) pathways. Differential peak analyses were based on the fold enrichment of peaks observed in different experiments. A peak was considered differential when the odds ratio between two groups was greater than 2. Utilizing the same methodology, genes associated with differential peaks were identified, and GO and KEGG enrichment analyses were performed on these genes as well.

## Statistics

Unless otherwise stated, all experiments were conducted with at least three independent replicates. The results are presented as the means ± standard deviations (SDs). Statistical analysis was carried out using Excel (Microsoft Office Home and Student 2019 version) and GraphPad Prism (version 8.02) for Microsoft Windows. The data were analyzed by two-sided t-test and are presented as mean values ± SD.

## Reporting summary

Further information on research design is available in the Nature Portfolio Reporting Summary linked to this article.

## Data availability

All datasets generated in this study have been deposited in the Gene Expression Omnibus repository under the series number GSE217831. Source data are provided with this paper.

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

## Acknowledgements

This work was funded by The National Key Research and Development Program of China (2017YFA0106600), the National Natural Science Foundation of China (No. 32070822, 82030023, 82322012, 31771609), The Key support object training project of Army Medical University (No. 2019R025), Young Doctoral Training Program of the Second Affiliated Hospital of Army Medical University (No. 2022YQB013 and 2022YQB060).

## Author contributions

C.L., J.Zhao, and Y.H. conceived the study; C.L. and X.L. designed the study; X.L., Z.H., J.Zhang, X.T., T.Y., Y.Z., S.L., F.L., Z.X., L.D., and W.Y. performed all the experiment; X.L., C.L., and J.Zhang carried out microscopy; C.L., Z.H., and X.L. prepared the manuscript; all authors approved the final version.

## Competing interests

The authors declare no competing interests.
