## [Peer Review File · Nature Communications]

Proenkephalin-A Secreted by Renal Proximal Tubules Functions as a Brake in Kidney RegenerationREVIEWER COMMENTS

Reviewer #1 (Remarks to the Author):

In this manuscript, Liu et al report novel insights into the mechanism of kidney regeneration in the zebrafish. The authors found that proenkephalin A (PENK-A) secreted by the renal proximal tubule is used to regulate regeneration. Specifically, they show that PENK-A inhibits the production of hydrogen peroxide, which the group previously found to be a key signal to stimulate regeneration following injury. However, the precise dosage and time release of hydrogen peroxide must be controlled, as high levels lead to unwelcome effects such as fibrosis. The authors here show that PENK-A is the crucial controller and acts upstream of changes in chromatin remodeling. The authors studies are quite thorough, with the implementation of complementary genetic studies and drug modulation of the relevant pathways. I found the study to be elegant, very important to expanding our understanding of neonephrogenesis following kidney injury, and thus overall poised to be impactful for the nephrology field. I look forward to future studies from this research group!

The comments below are intended to be constructive and helpful to the authors. As stated above, this work is outstanding.

Major comments

- 1) The authors do not discuss PENK-A in the introduction, and this would be a nice place to tell the reader more about this molecule.
- 2) In the figures, it is sometimes unclear whether the authors have RT-PCR data or both RT-PCR and qRT-PCR data to support their studies. In Figure 3A,B, and E, it would be important to have the qRT-PCR data to back up the authors' conclusions.
- 3) Figures 3K & 6H: it is very difficult to visualize the single cells: the manuscript would be improved by provision of additional images and/or insets to help readers see what the authors have noted in their research.
- 4) Can the authors please clarify their reasoning for the timing in Figures C & E, why not look at earlier time points?
- 5) Figure 6: Have the authors tested gain of function for tcf21?
- 6) Figure 6D: It would be more convincing to show a double fluorescent in situ or double

antibody staining for tcf21 in lhx1a+ cells.

Minor comments

7) Figure 2E,G: Use the same y-axis please. Why were 7 dpi examined, versus 5 dpi? Can the authors please expand their rationale?

8) Figure 7: Nice model! The word aggregation is misspelled (aggregation in the bottom left)

Reviewer #2 (Remarks to the Author):

Chi Lui et al. studied the role of Proenkephalin A in kidney regeneration following Acute Kidney Injury using zebrafish. They suggested that Proenkephalin A level, secreted by PTEC, is correlated with the loss of PTEC following chemically-triggered AKI, and inversely correlated with H₂O₂. In addition, they showed that Proenkephalin A-H₂O₂ pathway regulated the rate and termination of regeneration, most likely via tcf21 and , key player in renal progenitor cell aggregates formation.

Despite the new role of proenkephalin as a biomarker of kidney injury kidney injury in man, little is known of it's role in AKI. The studies presented by Chi Lui et al. are using a Zebrafish models of AKI during fish kidney development, which is not clearly discussed in the manuscript. This is an important aspect of this study which may limit its role to a developing kidney and not in a mature kidney, which would be more relevant to understand the role of proenkephalin in AKI in man. Furthermore, we question as to whether the AKI-regeneration concept used by the investigators in this manuscript is actually an AKI-in a developing kidney.

Manuscript is in a rather draft form than publication-ready and lack of scientific rigor (missing protocols, reagent or procedures or discrepancies between figures and results section/figure legend) required for publication.

Confirmatory experiments in human AKI kidney samples should strengthen the findings in ZF.

Major comments:

Fig 1:

How do the authors explain the partially overlapping of Met-ENK staining with EGFP (ie

proximal tubules staining) in Zebra fish? What is the rationale of studying penka only in EGFP^{high} cells and what is the level of penka in other lower expressing EGFP Proximal tubular cells?

The conclusion stating that “penka is specifically expressed in PTECs” seems premature. A co-staining using a PTEC marker (ie SLC20a1a for instance) could strengthen the data.

Knowing that zebrafish has 2 nephrons, the authors should elaborate how the nephrons were quantitated. The nephron quantitation method should be added in the M&M section to support the quantitation shown of the nephrons in Fig 1F. Distribution studies over time could be added for clarity in supplemental section.

Fig 2:

Have any off-target analyses of CrispR/Cas been assessed in penka mutants? This should be assessed to ensure the specificity of the penka KO performed by CrispR/Cas to make sure no other targets were hit in the mutant chosen.

The authors describe the “regeneration phenotype” in WT vs penka mutant. It is not clear as to whether this phenotype is assessed upon AKI caused with gentamycin or if this is a developmental assessment (dpi 3-15). This needs to be clarified. The statement that penka can negatively regulate the regeneration of nephrons is debatable, as the number of the RPCA (quantitated via *lhx1a*) could be related to the development state of the nephron in zebrafish.

How was 7 dpi time point was selected? It seems that 5 dpi (fig 2B) shows a greater differential than 7 dpi. What would be the effect of NAL-M passed the 7 dpi time-point, where *lhx1a* is greater in WT compared mutant. A time course including longer time points is necessary to understand if the number of aggregates is related to the level of mRNA.

Fig 3:

The authors used the *tg(hsp70l:penka)* to artificially increase penka, upon heat shock and show that the number of RPCA is decreased. How was the heat shock performed in the novel *tg(hsp70l:penka)* line? This information is lacking in the manuscript.

The authors then speculated that penka increase via penk-a agonists Met-ENK and TRAM, terminated regeneration prematurely. Have they attempted to inject penka mRNA in ZF to confirm that penka was responsible for kidney regeneration, or conversely have they

evaluated penka antisense-MO on kidney regeneration?

Is gentamycin used in these data? There is no mention of dose or treatment in results or in figure legend section. This needs to be addressed.

Fig 4:

Do panel C is the quantitation of images panel B, and panel E the quantitation of the images in panel D, Panel G is the quantitation of images in panel F? This should be made clearer in the results and in the figure legend sections.

The quality of the images presented in Fig B, D and F is poor. The nephron and the renal medulla should be delineated.

Line 211-212: The authors conclude that VAS2870 or duox1 vivo-MO inhibited kidney regeneration (Fig. 4G and H). Fig. 4G shows H₂O₂ measurement in WT and Tg(hsp70l:penka), ie up regulation of penka upon heat shock. Does this mean that heat shock was applied to the Tg(hsp70l:penka) mutant? Was heat shock also applied to WT as controls? The heat shock protocol is lacking and this needs to be clarified.

Fig 5:

Panel A: the legend is confusing. Genotypes should be used for each column instead of – or + (ie in sequence in the header: Penka +/+ -/- +/+ -/- +/+ -/- +/+ -/-).

What are RD3, RD5 and RD7? Please define the acronym in the Fig legend or use same nomenclature as the rest of the paper (3 dpi, 5 dpi or 7 dpi?).

Line 227-229: the authors state: “... the difference at 7 dpi was not significant compared to the control group (Fig. 5A, B).” the figure A and quantitation in B, shows a significant difference between WT and penka-/- . Please verify.

Line 234: the authors state: “After inhibition of H₂O₂ production by VAS2870 or duox1 vivo-MO, the decrease in H3K4me3 at 3 dpi was inhibited.” The authors should refer to the right panel in the figure (panel HIJ?). Also is this done in the context of a Gent-triggered AKI? This should be stated clearly in the results section.

Panel J: it looks like some white arrow heads are pointing at nothing. Please check.

Line 269-270. The authors state: “... we speculated that H₂O₂ promotes the formation of RPCAs by remodeling H3K4me3 modification in renal progenitor cells.” What kind of modification are they referring to? This statement is not supported by any experimental

data or literature. This needs to be addressed

Fig 6:

Panel A: the title is not clear. Do the authors mean “Analysis of the H3K4me3 binding pattern in the promoter region of tcf21 by ChIP seq”? please clarify.

Panel B: table X axis shows that 1, 3, 5, 7 and 9 dpi were investigated, the text mentioned 1-7 dpi. Please be consistent.

Again, it looks that the data presented in the Fig 6 are in the context of Gent-caused AKI as suggested in Fig E, F, G, K. This should be described in the text.

Lines 296-299. The authors state: “Furthermore, we examined the expression of tcf21 in the CPI-455 treated group and found that tcf21 expression did not increase after H3K4me3 remodeling was inhibited, which was consistent with the change in lhx1a expression (Fig. 6G).” Are the authors referring to the level of lhx1a upon CPI-455 shown in Fig. 5E? please clarify.

Fig. 6H: the contrast should be increased as the cells are difficult to see on the screen, let alone on a print.

Line 313: “... by increasing tcf21 promoter H3K4me3 modification and increasing its expression.” What are the modifications involved? This is confusing as no modifications of H3K4me3 are shown. Do the authors mean that H3K4me3 changes modified tcf21 promoter activity?

Minor:

Please make sure all abbreviations are stated: including but not limited to: BWA, KEGG

Figures section:

Fig 1F: please explain what kind of ration is described on Y axis: “The ratio of renal tubule (%)”. How is it quantitated and normalized to?

Fig 2B: please explain how the relative lhx1a level is normalized. Is it normalized to b-actin?

Fig 2C-E please mark with arrow what scientist should be looking (blue dots?).

Is Fig 2G the quantitation of data shown in Fig 2F? this should be clarified.

Methods section:

Please check grammar of this section but specifically:

Zebrafish

Line 380: The sentence doesn't make sense.

Generation and genotyping of penka mutants

Line 388: reagent is misspelled.

WISH

Line 413: Authors state "Briefly, fish with internal organs removed were fixed overnight in 4% paraformaldehyde (PFA)." but seem to contradict this statement (line 415) "... fixed kidneys were removed from body and permeabilized with proteinase K...). Were the kidneys left inside the cavity of the fish during fixation?

Immunofluorescence

Lines 422-424: The sentence "... and then embedded in OCT to receive the frozen section at 100 μm on a Micron HM550 cryostat" does not make sense.

FACS

Line 436-439: Please rephrase the sentence "To obtain the RNA library of interested cells, kidney cells of Tg(lhx1a:DsRed) or Tg(TSH β :EGFP) from 10 fishes were manually dissected in 1% PBS and 0.005 Trypsin-EDTA solution at 5 dpi. Interested cells were sorted using a MoFlo XDP (Beckman) and collected for RNA extraction. This paragraph is confusing.

Western blotting

Line 440-445: How was the quantity of the proteins normalized before loading on the gel?

Vivo-Morpholino and pharmaceutical treatment.

Lines 452-459: Please check grammar: During kidney regeneration, TRAM (16 μM , 10 μL per fish), NAL-455 M (2.5 μM , 10 μL per fish), Met-ENK (100 μM , 10 μL per fish), VAS2870 (8 μM , 10 μL per fish), and CPI-455 (80 μM , 10 μL per fish) was intraperitoneally injected into zebrafish at 2, 4, and 6 dpi. 0.1% DMSO was also applied as their control groups with the same conditions.

Line 461-462: cDNA are not "collected", may be the authors meant that the "RNA" are collected, please correct.

H2O2 measurement:

Line 465: the fish were injected with (missing "with").

Line 466: Did the authors mean: 3h post injection instead of "After 3 hours of injection"

suggesting that the injection lasted 3h?

Line 468: did the author meant: "The kidneys were "collected" and photographed "using a" the Nikon A1 confocal microscope".

Data availability:

Line 511-512: is this a duplication? "Source Data are included with this paper. Source data are provided with this paper."

We have made appropriate changes and careful revisions. In particular, we have performed fluorescent *in situ* hybridization (FISH) to examine the expression of *PENK* in human kidney samples. We found that *PENK* is expressed in the proximal tubules (PTs), similar to its expression in zebrafish. Furthermore, we observed a decrease in *PENK* expression during acute kidney injury (AKI) in humans, consistent with the findings in zebrafish. This suggests that a comprehensive understanding of *penka*'s impact on kidney regeneration in zebrafish could potentially provide valuable insights for the treatment of human kidney diseases. The revised parts have been highlighted in the manuscript and figure legends. Enclosed is our point-by-point response to the reviewers.

Responses to reviewers' comments:

Reviewer #1:

In this manuscript, Liu et al report novel insights into the mechanism of kidney regeneration in the zebrafish. The authors found that proenkephalin A (PENK-A) secreted by the renal proximal tubule is used to regulate regeneration. Specifically, they show that PENK-A inhibits the production of hydrogen peroxide, which the group previously found to be a key signal to stimulate regeneration following injury. However, the precise dosage and time release of hydrogen peroxide must be controlled, as high levels lead to unwelcome effects such as fibrosis. The authors here show that PENK-A is the crucial controller and acts upstream of changes in chromatin remodeling. The authors studies are quite thorough, with the implementation of complementary genetic studies and drug modulation of the relevant pathways. I found the study to be elegant, very important to expanding our understanding of neonephrogenesis following kidney injury, and thus overall poised to be impactful for the nephrology field. I look forward to future studies from this research group!

We sincerely appreciate your recognition of our research. Your feedback has greatly contributed to enhancing the quality of our paper. Thank you!

The comments below are intended to be constructive and helpful to the authors. As stated above, this work is outstanding.

Major comments

1) *The authors do not discuss PENK-A in the introduction, and this would be a nice place to tell the reader more about this molecule.*

Thank you for the suggestion. We have added a paragraph in the introduction to introduce PENK-A (lines 79-91).

2) *In the figures, it is sometimes unclear whether the authors have RT-PCR data or both RT-PCR and qRT-PCR data to support their studies. In Figure 3A, B, and E, it would be important to have the qRT-PCR data to back up the authors' conclusions.*

We apologize for any confusion caused by the lack of clarity regarding the type of data presented in the figures. To ensure transparency and support the conclusions more robustly, we have now included clear labeling in the manuscript and figure legends indicating whether the presented data is from RT-PCR or qRT-PCR experiments (**lines 140-141, 169, 179, 182, 193, 198, 206, etc**). Additionally, as you rightly pointed out, for Figures 3A, 3B, and 3E, where the conclusions needed stronger validation, we have included qRT-PCR data to back up our findings (**Fig. 3B, D, K**). Additionally, we have modified the method for assessing *tcf21* expression in Fig. 6 to qRT-PCR, providing a more precise representation of its changes (**Fig. 6E, F**).

3) *Figures 3K & 6H: it is very difficult to visualize the single cells: the manuscript would be improved by provision of additional images and/or insets to help readers see what the authors have noted in their research.*

We have replaced the original figures with new images that better illustrate the single cells and the observed cellular events. The new images will now allow readers to more easily visualize and interpret the data presented in **Fig. 3N and Fig. 7A**.

4) *Can the authors please clarify their reasoning for the timing in Figures C & E, why not look at earlier time points?*

In our previous approach, we focused mainly on investigating the dose-response relationship between H₂O₂ and *penka*, which led us to select only a few time points for analysis. However, based on your valuable suggestion, we now recognize the importance of including earlier time points to provide a more comprehensive view of the biological processes. We have introduced two new time points, 3 and 5 dpi (**Fig. 4C-G**). We also observed a dose-dependent effect of PENK-A in inhibiting H₂O₂ production at the newly added time points. We believe that the revised findings incorporating these new time points will offer a more complete understanding of the interplay between H₂O₂ and *penka* at different stages of the experimental process.

5) *Figure 6: Have the authors tested gain of function for tcf21?*

Thank you for your inquiry regarding Fig. 6. To better present the data and accommodate the inclusion of new images, we have divided the original Fig. 6 into two separate figures: **Fig. 6** and **Fig. 7**. In addition, we acknowledge that we did not perform gain-of-function experiments for *tcf21* in the initial version of the manuscript. Now, we created a transgenic line *Tg(hsp70l:tcf21)* that overexpresses *tcf21* effectively upon heat shock (**Fig. 7H**). Heat shocking at 2, 4 and 6 dpi resulted in a significant increase in the number of renal progenitor cell aggregates (RPCAs) in the injured kidneys of the *Tg(hsp70l:tcf21;lhx1a:DsRed)* fish, while the number of individual renal progenitor cells (RPCs) was decreased compared to that in the control group (**Fig. 7F, G**). RT-PCR and whole-mount *in situ* hybridization (WISH) showed increased *lhx1a* expression and RPCAs (**Fig. 7H-J**). Additionally, activating the PENK-A signaling pathway with intraperitoneal Met-ENK injection restored the number of RPCAs in the injured kidneys of heat-shocked *Tg(hsp70l:tcf21;lhx1a:DsRed)* fish (**Fig. 7F, G**). Confirmation was obtained through *lhx1a* RT-PCR and WISH (**Fig. 7H-J**). When treated with CPI-455 (80

μM , 10 μL per fish) in heat-shocked *Tg(hsp70l:pcf21;lhx1a:DsRed)* fish, it was found that the overexpression of *pcf21* could rescue the inhibition of RPC aggregation by CPI-455 (**Fig. 7F**) and reactivate kidney regeneration (**Fig. 7H-J**), supporting the role of PENK-A in regulating kidney regeneration through its effects on *pcf21*.

6) *Figure 6D: It would be more convincing to show a double fluorescent in situ or double antibody staining for pcf21 in lhx1a+ cells.*

Similar to *lhx1a*, *pax2a* is also a marker for RPCAs^{1, 2, 3}. In response to your comment, we employed a combined approach of *pcf21* FISH and Pax2a immunofluorescence, which enabled us to detect the expression of *pcf21* in RPCAs. The new results demonstrate the colocalization of *pcf21* expression in Pax2a⁺ RPCAs (**Fig. 6D**), further supporting our findings.

Minor comments

7) *Figure 2E,G: Use the same y-axis please. Why were 7 dpi examined, versus 5 dpi? Can the authors please expand their rationale?*

Thank you for your suggestion. We have revised Figure 2E and G to use the same y-axis scale (**Fig. 2D, G**). The reason we chose 7 dpi is that it is usually the time point when RPCAs are most abundant and thus represents the overall kidney regeneration process effectively. However, based on the suggestions from you and the other reviewer, we have added two additional time points of observation, 5 dpi and 9 dpi (**Fig. 2C-G**). Interestingly, we found that similar to the *penka* mutants, the zebrafish treated with NAL-M also exhibited accelerated kidney regeneration that peaked at 5 and 7 dpi and then declined at 9 dpi. Additionally, regarding the expression changes of *pcf21* in the *penka* mutants, we have also included the time point of 5 dpi to better capture the differences between the mutants and WT (**Fig. 6E**). In the case of other experiments involving WT kidneys, the decision to focus on the 7 dpi time point is attributed to the relatively insignificant RPACs production by WT kidneys at 5 dpi. By opting for 7 dpi, we can effectively showcase the PENK-A related signaling pathway's impact on kidney regeneration.

8) *Figure 7: Nice model! The word aggregation is misspelled (aggregation in the bottom left).*

We apologize for the oversight. In the revised version of the manuscript, we have corrected the misspelling of "aggregation" in the figure (**Fig. 8**).

Reviewer #2:

Chi Lui et al. studied the role of Proenkephalin A in kidney regeneration following Acute Kidney Injury using zebrafish. They suggested that Proenkephalin A level, secreted by PTEC, is correlated with the loss of PTEC following chemically-triggered AKI, and inversely correlated with H2O2. In addition, they showed that Proenkephalin A-H2O2 pathway regulated the rate

and termination of regeneration, most likely via tcf21 and , key player in renal progenitor cell aggregates formation.

Despite the new role of proenkephalin as a biomarker of kidney injury kidney injury in man, little is known of it's role in AKI. The studies presented by Chi Lui et al. are using a Zebrafish models of AKI during fish kidney development, which is not clearly discussed in the manuscript. This is an important aspect of this study which may limit its role to a developing kidney and not in a mature kidney, which would be more relevant to understand the role of proenkephalin in AKI in man. Furthermore, we question as to whether the AKI-regeneration concept used by the investigators in this manuscript is actually an AKI-in a developing kidney.

Manuscript is in a rather draft form than publication-ready and lack of scientific rigor (missing protocols, reagent or procedures or discrepancies between figures and results section/figure legend) required for publication.

Confirmatory experiments in human AKI kidney samples should strengthen the findings in ZF.

Thank you for your comprehensive evaluation and valuable suggestions. We deeply regret any language expression and organizational issues that might have been present in our previous manuscript. To enhance clarity, we have carried out substantial revisions. The Methods section has been extensively reworked to provide detailed insights into the experimental techniques employed in our study. The information regarding reagents or procedures is listed in **Table S2**. Additional improvements have been incorporated into the figure legends, highlighting the relationships among the images. Furthermore, we have addressed inaccuracies in the results section related to image references. To enhance the manuscript's quality, we have enlisted the assistance of English language editing services to make further improvements.

Most importantly, we deeply apologize for any confusion caused by the lack of explicit clarification regarding our focus on adult zebrafish kidneys. In the revised manuscript, we have made it explicitly clear that all experiments related to kidney injury were exclusively conducted using adult zebrafish kidneys (**lines 138-139**). Therefore, it is crucial to note that the experimental results in this study were uninfluenced by kidney development. Following your suggestion, we utilized FISH to examine changes in *PENK* expression in human kidney samples. Remarkably, we found that human *PENK*, much like its zebrafish counterpart, is also expressed in the PTs and exhibits significant downregulation in the kidneys of AKI patients (**Fig. 1D**). These findings underscore the potential functional role of *PENK* in human AKI pathology. Thus, the utilization of zebrafish to explore the function of *penka* in kidney regeneration holds promise for advancing AKI treatment strategies.

Major comments:

Fig 1:

How do the authors explain the partially overlapping of Met-ENK staining with EGFP (ie proximal tubules staining) in Zebrafish? What is the rationale of studying penka only in EGFP^{high} cells and what is the level of penka in other lower expressing EGFP Proximal tubular cells?

The conclusion stating that “penka is specifically expressed in PTECs” seems premature. A co-staining using a PTEC marker (ie SLC20a1a for instance) could strengthen the data.

The partial colocalization of Met-ENK staining and EGFP may be attributable to *Tg(gtsh β :GFP)* not fully marking the PTs. To address this issue, we first performed immunostaining using a kidney tubule marker Pax2a and combined it with *penka* FISH. By employing this method, we confirmed the expression of *penka* in the kidney tubules (**Fig. 1B**). As the PT comprises both the proximal convoluted tubule (PCT) and proximal straight tubule (PST), we used FISH to label the PCTs with the marker gene *slc20a1a* and PSTs with the marker gene *trmp7* to comprehensively mark the PTs⁴. By combining Met-ENK immunostaining with *slc20a1a* and *trmp7* FISH, we further confirmed the specific expression of *penka* in the PTs (**Fig. 1C**). In addition, we also conducted FISH to detect *PENK* expression in human patient kidney specimens. We found that *PENK* was expressed in the PTs, as indicated by colocalization with lotus tetragonolobus lectin (**Fig. 1D**), a marker for PTs. Moreover, following the occurrence of AKI, *PENK* expression was significantly decreased in human kidneys (**Fig. 1D**). Upon integrating these findings with single-cell sequencing data from zebrafish kidneys (**Fig. 1A**), we can confidently establish that *penka* is specifically expressed in the PTs.

Knowing that zebrafish has 2 nephrons, the authors should elaborate how the nephrons were quantitated. The nephron quantitation method should be added in the M&M section to support the quantitation shown of the nephrons in Fig 1F. Distribution studies over time could be added for clarity in supplemental section.

We apologize for any confusion caused by not clearly specifying in the initial manuscript that our study focused on adult zebrafish kidneys. In the revised version, we have clarified that all experiments involving kidney injury were conducted using adult zebrafish kidneys (**lines 138, 139**). Zebrafish kidney development includes two stages, the pronephros and the mesonephros stages. During the embryonic stage, the pronephros of zebrafish consists of two nephrons, while the adult zebrafish mesonephros contains approximately 500 nephrons¹. We have also added this background information in the introduction section (**lines 53-56**). To accurately count the nephrons, we utilized the *Tg(gtsh β :GFP)* transgenic line, in which a subset of PTs is specifically marked⁵. To perform the counting, kidneys were carefully removed from the adult *Tg(gtsh β :GFP)* zebrafish and imaged in a 1.5 mm by 1.5 millimeter area using a Nikon A1 confocal microscope. The total number of GFP-positive segments (corresponding to the number of nephrons) was determined using ImageJ. To assess the variation in the number of nephrons following Gent-induced AKI, the baseline number of nephrons in the uninjured kidneys was considered 100%. Subsequently, the percentage change in nephrons after AKI compared to the uninjured condition was utilized to depict the extent of kidney

damage (**Supplementary Fig. S1**). This method has been further explained in detail in the Methods section of the revised manuscript (**lines 509-522**).

Fig 2:

Have any off-target analyses of CrispR/Cas been assessed in penka mutants? This should be assessed to ensure the specificity of the penka KO performed by CrispR/Cas to make sure no other targets were hit in the mutant chosen.

Previous studies have indicated that the probability of CRISPR off-target effects in zebrafish is relatively low, particularly for stable mutant strains^{6, 7}. The off-target mutation rates have been reported to range from 3.17% to 0.07% in zebrafish, based on sequencing of the top three to four predicted off-target regions using sequence homology^{8, 9, 10}. The *penka*⁻¹⁺²⁴ mutant strains we used were mainly derived from F3 generations, and the probability that they contained off-target mutations was low. However, to confirm that our *penka*⁻¹⁺²⁴ mutant strains were indeed free from off-target effects, we employed CRISPRScan to predict potential off-target sites for the gRNA used in our study^{11, 12}. Additionally, we performed sequencing on the top five predicted off-target sites to check for any mutations. The results showed that, compared to the WT, the mutant strains did not exhibit any mutations at these five sites (**lines 163-167; Supplementary Fig. S2D, E**). These findings strongly suggest that our mutant strains were free from off-target effects.

*The authors describe the “regeneration phenotype” in WT vs penka mutant. It is not clear as to whether this phenotype is assessed upon AKI caused with gentamycin or if this is a developmental assessment (dpi 3-15). This needs to be clarified. The statement that penka can negatively regulate the regeneration of nephrons is debatable, as the number of the RPCA (quantitated via *lhx1a*) could be related to the development state of the nephron in zebrafish.*

We apologize once again for any lack of clarity regarding the kidney injury model in our manuscript. We want to reiterate that all kidney injury experiments were indeed conducted using intraperitoneal injection of gentamicin in adult zebrafish. In the revised manuscript, we have made a clear, explicit statement that all kidney injury experiments in this study were carried out using this method (**lines 138, 139**). Therefore, all the observed "regeneration phenotypes" in this study occurred in the context of the adult kidneys and were not related to kidney development.

*How was 7 dpi time point was selected? It seems that 5 dpi (fig 2B) shows a greater differential than 7 dpi. What would be the effect of NAL-M passed the 7 dpi time-point, where *lhx1a* is greater in WT compared mutant. A time course including longer time points is necessary to understand if the number of aggregates is related to the level of mRNA.*

Thank you for your thoughtful questions and observations regarding the time points in our study. The reason we chose 7 dpi was to capture the time point when the number of regenerated RPCAs is at its peak, providing a comprehensive reflection of the overall kidney regeneration process. In other zebrafish kidney regeneration studies, 7 dpi is commonly selected to assess the status of kidney regeneration^{2, 13, 14, 15}. We also greatly appreciate your perspective, as we agree that the differences between the mutants and WT are most pronounced at 5 dpi. Adding this time point will better reflect the function of *penka*. To provide a more comprehensive evaluation of the impact of NAL-M on kidney regeneration, we included additional time points at 5 dpi and 9 dpi (**Fig. 2C-G**). Our findings revealed that, similar to the *penka* mutation, NAL-M treatment also accelerates kidney regeneration. At 9 dpi, WISH analysis demonstrated reductions in the numbers of *lhx1a*⁺ RPCAs in the *penka* mutant and NAL-M-treated groups compared to the control group (**Fig. 2C-D, F-G**), confirming a correlation between the number of RPCAs and the level of *lhx1a* mRNA. These data indicate that the inhibition of PENK-A signaling could accelerate kidney regeneration. Additionally, regarding the expression changes of *tcf21* in the *penka* mutant kidneys, we have also included the time point of 5 dpi to better capture the differences between the mutants and WT (**Fig. 6E**). In the case of other experiments involving WT kidneys, the decision to focus on the 7 dpi time point is attributed to the relatively insignificant RPACs production by WT kidneys at 5 dpi. By opting for 7 dpi, we can effectively showcase the PENK-A related signaling pathway's impact on kidney regeneration.

Fig 3:

The authors used the tg(hsp70l:penka) to artificially increase penka, upon heat shock and show that the number of RPAC is decreased. How was the heat shock performed in the novel tg(hsp70l:penka) line? This information is lacking in the manuscript.

Thank you for your reminder! We have included a description of the heat shock treatment for adult zebrafish in the revised manuscript (**lines 629-633**). Here is the revised description: "For heat shock treatment, zebrafish were transferred to preheated system water at 39°C and kept at 39°C for 30 minutes. After heat shock, the zebrafish were returned to the 28.5°C system water. Zebrafish were heat shocked every other day after 2 dpi until the kidneys were collected for subsequent experiments."

The authors then speculated that penka increase via penk-a agonists Met-ENK and TRAM, terminated regeneration prematurely. Have they attempted to inject penka mRNA in ZF to confirm that penka was responsible for kidney regeneration, or conversely have they evaluated penka antisense-MO on kidney regeneration?

In adult zebrafish, there is currently no effective method for direct mRNA delivery. Therefore, to achieve gain-of-function experiments, we utilized the *Tg(hsp70l:penka)* transgenic line for overexpression. Furthermore, following

your advice, we included *penka* vivo-MO experiments (**Supplementary Fig. S3**). By administering *penka* vivo-MO through intraperitoneal injection, we achieved effective knockdown of *penka* mRNA in the kidneys (**Supplementary Fig. S3A, B**). Remarkably, we observed that after *penka* knockdown, kidney regeneration was significantly accelerated, as demonstrated by *lhx1a* RT-PCR and WISH analyses (**Supplementary Fig. S3C-E**), which was consistent with the observed phenotype in *penka* mutant kidneys.

Is gentamycin used in these data? There is no mention of dose or treatment in results or in figure legend section. This needs to be addressed.

As mentioned in the previous response, we have emphasized in both the manuscript and figure legends that the adult zebrafish kidney injury model was achieved through intraperitoneal injection of gentamicin (2.7 µg/µL, 20 µL per fish) (**lines 135-139, 946, 955, 997**). This approach was used consistently throughout our study to assess kidney regeneration in adult zebrafish. We apologize for any previous oversight and have now made the necessary clarifications in the revised manuscript to accurately describe the use of adult zebrafish with gentamicin-induced kidney injury.

Fig 4:

Do panel C is the quantitation of images panel B, and panel E the quantitation of the images in panel D, Panel G is the quantitation of images in panel F? This should be made clearer in the results and in the figure legend sections.

Thank you for your feedback. We have incorporated the necessary revisions into the revised manuscript to enhance the clarity of relationships between the panels in the figures and their respective quantitation (**lines 957, 964, 972, 985, 1020, 1036-1039, etc**). Furthermore, for precise measurement of H₂O₂ production, we have modified the previous method of PBSF staining and adopted a fluorimetric hydrogen peroxide assay kit. As a result, we have substituted the original images in Fig. 4B-G, S4 with the new **Fig. 4C-G**.

The quality of the images presented in Fig B, D and F is poor. The nephron and the renal medulla should be delineated.

To better distinguish renal tubules from the renal medulla, we utilized a transgenic line *Tg(cdh17:DsRed)* with specific labeling of renal tubules with DsRed². Utilizing this transgenic line, we observed that after 3 dpi, the generation of H₂O₂ was concentrated primarily in the renal medulla (**Fig. 4B**). Additionally, to precisely measure H₂O₂ production, we modified the previous method of PBSF staining and used a fluorimetric hydrogen peroxide assay kit. This modification minimized errors arising from image quality (**Fig. 4C-G**). Employing this assay kit, we arrived at the same conclusion as in the original manuscript: *penka* can exert precise control over H₂O₂ production in the kidneys, thereby influencing kidney regeneration.

Line 211-212: The authors conclude that VAS2870 or duox1 vivo-MO inhibited kidney regeneration (Fig. 4G and H). Fig. 4G shows H₂O₂ measurement in WT and *Tg(hsp70l:penka)*, ie up regulation of penka upon heat shock. Does this mean that heat shock was applied to the *Tg(hsp70l:penka)* mutant? Was heat shock also applied to WT as controls? The heat shock protocol is lacking and this needs to be clarified.

We apologize for the error in the figure reference. The correct reference in the original manuscript should be Fig. 4H and I, not Fig. 4G and H. We have made the necessary corrections in the revised manuscript (**lines 269-271; Fig. 4H-J**). Thank you for bringing this to our attention. Therefore, we did not employ the *Tg(hsp70l:penka)* mutant in our study. In the original Fig. 4G, the WT control was also subjected to heat shock treatment, and this has been explicitly labeled in the new **Fig. 4G**. Additionally, the heat shock protocol has been supplemented in the Methods section for clarification (**lines 629-633**).

Fig 5:

Panel A: the legend is confusing. Genotypes should be used for each column instead of – or + (ie in sequence in the header: Penka +/+ -/- +/+ -/- +/+ -/- +/+ -/-).

What are RD3, RD5 and RD7? Please define the acronym in the Fig legend or use same nomenclature as the rest of the paper (3 dpi, 5 dpi or 7 dpi?).

Thank you for your suggestions. "RD3," "RD5," and "RD7" are abbreviations for Regeneration Day (RD) 3, 5, and 7, which previously referred to the respective days of kidney regeneration after injury. These terms are equivalent to "3 dpi," "5 dpi," and "7 dpi," denoting the same time points after kidney injury. To ensure clarity, we have uniformly used "dpi" throughout this revision to indicate the temporal context, instead of using "RD." We have also incorporated these changes in the revised figure (**Fig. 5A**).

Line 227-229: the authors state: "... the difference at 7 dpi was not significant compared to the control group (Fig. 5A, B)." the figure A and quantitation in B, shows a significant difference between WT and *penka*^{-/-}. Please verify.

We apologize for the confusion caused by our previous description. Here, we are referring to the changes in H3K4me3 levels in WT zebrafish, not in the *penka* mutant. In the revised manuscript, we have modified the wording as follows: " Furthermore, in WT kidneys, a subsequent increase in H3K4me3 levels was observed at 5 dpi, and by 7 dpi, the difference from the levels in the uninjured group was not statistically significant (**Fig. 5A, B**). However, at 5 and 7 dpi, the H3K4me3 levels in *penka*⁻¹⁺²⁴ mutant kidneys were lower than those in WT kidneys (**Fig. 5A, B**) (**lines 292-296**)."

Line 234: the authors state: “After inhibition of H₂O₂ production by VAS2870 or duox1 vivo-MO, the decrease in H3K4me3 at 3 dpi was inhibited.” The authors should refer to the right panel in the figure (panel HIJ?). Also is this done in the context of a Gent-triggered AKI? This should be stated clearly in the results section.

We sincerely apologize for the oversight regarding the missing image references in this section. In the revised manuscript, we have rectified this by including the appropriate image references (**lines 302-304; Fig. 5C, D**). Your reminder is greatly appreciated. Additionally, we have enhanced the clarity by specifying that all kidney injury experiments in this study were exclusively carried out using the gentamicin intraperitoneal injection method in adult zebrafish (**lines 138, 139**).

Panel J: it looks like some white arrow heads are pointing at nothing. Please check.

We have replaced the images with new ones to allow readers to clearly distinguish individual RPCs (**Fig. 5J**).

Line 269-270. The authors state: “... we speculated that H₂O₂ promotes the formation of RPCAs by remodeling H3K4me3 modification in renal progenitor cells.” What kind of modification are they referring to? This statement is not supported by any experimental data or literature. This needs to be addressed

Thank you for bringing this concern to our attention. We apologize for the lack of clarity in the statement made in lines 269-270. As per your suggestion, we have revised the description in the manuscript. H3K4me3 is defined as the trimethylation of the 4th lysine residue of the histone H3 protein, occurring at the promoter region and being linked to the activation of nearby gene expression (**lines 103-105, 281-284**). Additionally, we have clarified that H₂O₂ influences the changes in H3K4me3 levels at the promoter regions of genes crucial for the RPC MET process (**lines 331-334**). The specific modifications are as follows: “We speculate that H₂O₂ may activate the expression of genes crucial for the RPC MET process by increasing H3K4me3 levels in the promoter regions of these genes, ultimately leading to the promotion of RPCA formation.” In our subsequent research, we found significant changes in the H3K4me3 levels at the *tcf21* promoter region through the analysis of ChIP-seq data (**Fig. 6A**), thus confirming this hypothesis.

Fig 6:

Panel A: the title is not clear. Do the authors mean “Analysis of the H3K4me3 binding pattern in the promoter region of tcf21 by ChIP seq”? please clarify.

Thank you for the correction. We have made the necessary revisions according to your suggestions (**lines 1052, 1053**).

Panel B: table X axis shows that 1, 3, 5, 7 and 9 dpi were investigated, the text mentioned 1-7 dpi. Please be consistent.

Thank you for the correction. We have made the necessary revisions according to your suggestions (**line 1055**).

Again, it looks that the data presented in the Fig 6 are in the context of Gent-caused AKI as suggested in Fig E, F, G, K. This should be described in the text.

Thank you for the reminder. In the revised manuscript, we have clarified that all kidney injury experiments in this study were conducted using the gentamicin intraperitoneal injection method in adult zebrafish (**lines 138, 139**).

Lines 296-299. The authors state: "Furthermore, we examined the expression of tcf21 in the CPI-455 treated group and found that tcf21 expression did not increase after H3K4me3 remodeling was inhibited, which was consistent with the change in lhx1a expression (Fig. 6G)." Are the authors referring to the level of lhx1a upon CPI-455 shown in Fig. 5E? please clarify.

Yes. To provide a clearer representation of this relationship, we have added reference to Figure 5E in the revised manuscript (**lines 368-369; Fig. 5G**). Thank you for the reminder!

Fig. 6H: the contrast should be increased as the cells are difficult to see on the screen, let alone on a print.

We have replaced these images to make the individual RPCs clearly visible (**Fig. 7A**).

Line 313: "... by increasing tcf21 promoter H3K4me3 modification and increasing its expression." What are the modifications involved? This is confusing as no modifications of H3K4me3 are shown. Do the authors mean that H3K4me3 changes modified tcf21 promoter activity?

We apologize for any confusion caused. Yes, here we refer to the changes in H3K4me3 levels at the *tcf21* promoter region, which will impact the activity of the *tcf21* promoter and alter the gene's expression. We have made the necessary changes in the original text based on your suggestions (**lines 397-400**). Additionally, we have flipped the orientation of the image in Figure 6A to help readers quickly locate the promoter region (**Fig. 6A**). From the images, one can observe a decrease in H3K4me3 levels at the *tcf21* promoter region at 3 dpi, followed by an increase at 5 dpi, which aligns with the activation timing of *tcf21* (**Fig. 6B**).

Minor:

Please make sure all abbreviations are stated: including but not limited to: BWA, KEGG

We have made the necessary revisions in the revised manuscript. Thank you for your valuable suggestions (**lines 682, 689, 691, 692, 700**).

Figures section:

Fig 1F: please explain what kind of ration is described on Y axis: “The ratio of renal tubule (%)”. How is it quantitated and normalized to?

To perform the counting, kidneys were carefully removed from the adult *Tg(gtsh β :GFP)* fish and imaged in a 1.5 mm by 1.5 millimeter area using a Nikon A1 confocal microscope. The total number of GFP positive segments was determined using ImageJ. To assess the variation in the number of nephrons following Gent-induced AKI, the baseline number of nephrons in the uninjured kidneys was considered 100%. Subsequently, the percentage change in nephrons after AKI compared to the uninjured condition was utilized to depict the extent of kidney damage. We have provided further elaboration in the Methods section (**lines 509-522**).

*Fig 2B: please explain how the relative *lhx1a* level is normalized. Is it normalized to *b-actin*?*

Yes, it was normalized to β -actin, and we have included this information in the figure legends (**lines 956, 957**).

Fig 2C-E please mark with arrow what scientist should be looking (blue dots?).

Yes, the blue dots represent cell clusters. We have also added new explanations in the figure legends (**line 961**).

Is Fig 2G the quantitation of data shown in Fig 2F? this should be clarified.

Thank you for the reminder. We have modified the relevant legends (**lines 963-964**).

Methods section:

Please check grammar of this section but specifically:

We have rewritten the Methods section to make it clearer and more comprehensible. Thank you for your detailed suggestions.

Zebrafish

Line 380: The sentence doesn't make sense.

We have corrected these errors in the revised manuscript, Thank you very much **(lines 464-465)**.

Generation and genotyping of penka mutants

Line 388: reagent is misspelled.

We have rewritten this paragraph **(lines 523-546)**. Thank you very much.

WISH

Line 413: Authors state "Briefly, fish with internal organs removed were fixed overnight in 4% paraformaldehyde (PFA)." but seem to contradict this statement (line 415) "... fixed kidneys were removed from body and permeabilized with proteinase K...). Were the kidneys left inside the cavity of the fish during fixation?"

Yes, we fixed the entire body with PFA after removing the abdominal organs except the kidneys, and then we extracted the fixed kidneys. We have made revisions in the revised manuscript to clarify this part **(lines 548-551)**. Thank you for your valuable feedback.

Immunofluorescence

Lines 422-424: The sentence "... and then embedded in OCT to receive the frozen section at 100 μm on a Micron HM550 cryostat" does not make sense.

We have rewritten this part **(lines 559, 560)**. Thank you very much.

FACS

Line 436-439: Please rephrase the sentence "To obtain the RNA library of interested cells, kidney cells of Tg(Ihx1a:DsRed) or Tg(TSHβ:EGFP) from 10 fishes were manually dissected in 1% PBS and 0.005 Trypsin-EDTA solution at 5 dpi. Interested cells were sorted using a MoFlo XDP (Beckman) and collected for RNA extraction. This paragraph is confusing.

We have made revisions to this section to clearly express our intent **(lines 659-662)**.

Western blotting

Line 440-445: How was the quantity of the proteins normalized before loading on the gel?

Thank you for the question. The protein concentration was determined using bicinchoninic acid protein assay kits (CWBIO, CW0014S). We have added this information in the Methods section (**lines 583, 584**).

Vivo-Morpholino and pharmaceutical treatment.

Lines 452-459: Please check grammar: During kidney regeneration, TRAM (16 μ M, 10 μ L per fish), NAL-455 M (2.5 μ M, 10 μ L per fish), Met-ENK (100 μ M, 10 μ L per fish), VAS2870 (8 μ M, 10 μ L per fish), and CPI-455 (80 μ M, 10 μ L per fish) was intraperitoneally injected into zebrafish at 2, 4, and 6 dpi. 0.1% DMSO was also applied as their control groups with the same conditions.

We have corrected the errors in the revised manuscript (**lines 589-593**). Thank you very much.

Line 461-462: cDNA are not “collected”, may be the authors meant that the “RNA” are collected, please correct.

We have corrected this error in the revised manuscript (**line 572-574**). Thank you very much.

H2O2 measurement:

Line 465: the fish were injected with (missing “with”).

We have corrected this error in the revised manuscript (**line 643**). Thank you very much.

Line 466: Did the authors mean: 3h post injection instead of “After 3 hours of injection” suggesting that the injection lasted 3h?

We have corrected this error in the revised manuscript (**lines 644, 645**). Thank you very much.

Line 468: did the author meant: “The kidneys were “collected” and photographed “using a” the Nikon A1 confocal microscope”.

We have corrected the errors in the revised manuscript (**lines 646-647**). Thank you very much.

Data availability:

Line 511-512: is this a duplication? “Source Data are included with this paper. Source data are provided with this paper.”

We have corrected this error in the revised manuscript (lines 705, 706). Thank you very much.

References

1. Diep CQ, *et al.* Identification of adult nephron progenitors capable of kidney regeneration in zebrafish. *Nature* **470**, 95-100 (2011).
2. Liu X, *et al.* Renal interstitial cells promote nephron regeneration by secreting prostaglandin E2. *Elife* **12**, (2023).
3. Chen J, *et al.* Dual roles of hydrogen peroxide in promoting zebrafish renal repair and regeneration. *Biochem Biophys Res Commun* **516**, 680-685 (2019).
4. Poureetezadi SJ, Cheng CN, Chambers JM, Drummond BE, Wingert RA. Prostaglandin signaling regulates nephron segment patterning of renal progenitors during zebrafish kidney development. *Elife* **5**, (2016).
5. Wang Y, Sun ZH, Zhou L, Li Z, Gui JF. Grouper tshbeta promoter-driven transgenic zebrafish marks proximal kidney tubule development. *PLoS One* **9**, e97806 (2014).
6. Varshney GK, *et al.* A high-throughput functional genomics workflow based on CRISPR/Cas9-mediated targeted mutagenesis in zebrafish. *Nat Protoc* **11**, 2357-2375 (2016).
7. Liu K, Petree C, Requena T, Varshney P, Varshney GK. Expanding the CRISPR Toolbox in Zebrafish for Studying Development and Disease. *Front Cell Dev Biol* **7**, 13 (2019).
8. Hruscha A, *et al.* Efficient CRISPR/Cas9 genome editing with low off-target effects in zebrafish. *Development* **140**, 4982-4987 (2013).
9. Burger A, *et al.* Maximizing mutagenesis with solubilized CRISPR-Cas9 ribonucleoprotein complexes. *Development* **143**, 2025-2037 (2016).
10. Shah AN, Davey CF, Whitebirch AC, Miller AC, Moens CB. Rapid reverse genetic screening using CRISPR in zebrafish. *Nat Methods* **12**, 535-540 (2015).
11. Moreno-Mateos MA, *et al.* CRISPRscan: designing highly efficient sgRNAs for CRISPR-Cas9 targeting in vivo. *Nat Methods* **12**, 982-988 (2015).

12. Uribe-Salazar JM, Kaya G, Sekar A, Weyenberg K, Ingamells C, Dennis MY. Evaluation of CRISPR gene-editing tools in zebrafish. *BMC Genomics* **23**, 12 (2022).
13. Kamei CN, Gallegos TF, Liu Y, Hukriede N, Drummond IA. Wnt signaling mediates new nephron formation during zebrafish kidney regeneration. *Development* **146**, (2019).
14. Gallegos TF, Kamei CN, Rohly M, Drummond IA. Fibroblast growth factor signaling mediates progenitor cell aggregation and nephron regeneration in the adult zebrafish kidney. *Dev Biol* **454**, 44-51 (2019).
15. McCampbell KK, Springer KN, Wingert RA. Atlas of Cellular Dynamics during Zebrafish Adult Kidney Regeneration. *Stem Cells Int* **2015**, 547636 (2015).

REVIEWERS' COMMENTS

Reviewer #1 (Remarks to the Author):

The authors have prepared an excellent revision and nicely utilized suggestions from the reviewers. I endorse the manuscript for publication.

Reviewer #3 (Remarks to the Author):

I appreciate the authors' dedication to improving the article. The study found that proenkephalin-A (PENK-A) expressed by renal proximal tubular epithelial cells acts as an inhibitor in kidney regeneration by reducing hydrogen peroxide (H₂O₂) production in a dose-dependent manner. The study utilized penka mutants and pharmaceutical treatments to demonstrate the involvement of PENK-A in regulating the rate and termination of regeneration. Additionally, the study revealed that H₂O₂ affects the expression of tcf21, a crucial factor in the formation of renal progenitor cell aggregates, by remodeling H3K4me3 in renal cells.

Responses to reviewers' comments:

Reviewer #1 (Remarks to the Author):

The authors have prepared an excellent revision and nicely utilized suggestions from the reviewers. I endorse the manuscript for publication.

Thank you for your affirmation. Your feedback has greatly contributed to improving the quality of our manuscript.

Reviewer #3 (Remarks to the Author):

I appreciate the authors' dedication to improving the article. The study found that proenkephalin-A (PENK-A) expressed by renal proximal tubular epithelial cells acts as an inhibitor in kidney regeneration by reducing hydrogen peroxide (H₂O₂) production in a dose-dependent manner. The study utilized penka mutants and pharmaceutical treatments to demonstrate the involvement of PENK-A in regulating the rate and termination of regeneration. Additionally, the study revealed that H₂O₂ affects the expression of tcf21, a crucial factor in the formation of renal progenitor cell aggregates, by remodeling H3K4me3 in renal cells.

Thank you for your comments and feedback. Your input has been instrumental in improving the quality of our manuscript.